# Fully Heteroscedastic Count Regression with Deep Double Poisson Networks

**Spencer Young** [* 1]  **Porter Jenkins** [* 2]  **Longchao Da** [3]  **Jeffrey Dotson** [4]  **Hua Wei** [3]

## Abstract

Neural networks capable of accurate, input-conditional uncertainty representation are essential for real-world AI systems. Deep ensembles of Gaussian networks have proven highly effective for continuous regression due to their ability to flexibly represent aleatoric uncertainty via unrestricted heteroscedastic variance, which in turn enables accurate epistemic uncertainty estimation. However, no analogous approach exists for *count* regression, despite many important applications. To address this gap, we propose the Deep Double Poisson Network (DDPN), a novel neural discrete count regression model that outputs the parameters of the Double Poisson distribution, enabling arbitrarily high or low predictive aleatoric uncertainty for count data and improving epistemic uncertainty estimation when ensembled. We formalize and prove that DDPN exhibits robust regression properties similar to heteroscedastic Gaussian models via learnable loss attenuation, and introduce a simple loss modification to control this behavior. Experiments on diverse datasets demonstrate that DDPN outperforms current baselines in accuracy, calibration, and out-of-distribution detection, establishing a new state-of-the-art in deep count regression.

## 1. Introduction

The pursuit of neural networks capable of learning accurate and reliable uncertainty representations has gained significant traction in recent years (Lakshminarayanan et al., 2017; Kendall & Gal, 2017; Gawlikowski et al., 2023; Dheur & Taieb, 2023). Input-dependent uncertainty is useful for

---
[*]Equal contribution  [1]Delicious AI [2]Brigham Young University, Department of Computer Science [3]Arizona State University, School of Computing and Augmented Intelligence [4]Ohio State University, Department of Marketing. Correspondence to: Spencer Young <spencer.young@deliciousai.com>, Porter Jenkins <pjenkins@cs.byu.edu>.

*Proceedings of the 42$^{nd}$ International Conference on Machine Learning*, Vancouver, Canada. PMLR 267, 2025. Copyright 2025 by the author(s).

detecting out-of-distribution data (Amini et al., 2020; Liu et al., 2020; Kang et al., 2023), active learning (Settles, 2009; Ziatdinov, 2024), reinforcement learning (Yu et al., 2020; Jenkins et al., 2022), and real-world decision-making under uncertainty (Abdar et al., 2021). While uncertainty quantification applied to regression on continuous outputs is well-studied, training neural networks to make probabilistic predictions over discrete counts has traditionally received less attention, despite multiple relevant applications. In recent years, neural networks have been trained to predict the size of crowds (Zhang et al., 2016; Lian et al., 2019; Zou et al., 2019; Luo et al., 2020; Zhang & Chan, 2020; Lin & Chan, 2023), the number of cars in a parking lot (Hsieh et al., 2017), traffic flow (Lv et al., 2014; Li et al., 2020; Liu et al., 2021), agricultural yields (You et al., 2017), inventory of product on shelves (Jenkins et al., 2023), and bacteria in microscopic images (Marsden et al., 2018).

Uncertainty is often decomposed into two quantities: *epistemic*, which refers to uncertainty due to misidentification of model parameters, and *aleatoric*, which is uncertainty due to observation noise (Der Kiureghian & Ditlevsen, 2009). In regression tasks, a popular and effective approach for capturing epistemic uncertainty is to use deep ensembles (DEs) (Lakshminarayanan et al., 2017) where a set of $M$ neural networks are independently trained from different initializations and combined to make predictions. Aleatoric uncertainty, meanwhile, is accounted for in each individual member, which outputs a distinct predictive distribution over the target. Aleatoric uncertainty can be categorized as *homoscedastic*, where the predictive uncertainty is constant for all inputs, and *heteroscedastic*, where predictive uncertainty varies as a function of the input. Existing work on DEs for regression trains each member of the ensemble to predict the mean and variance of a Gaussian distribution, $\left[\mu_m^{(i)}, \sigma_m^{2\,(i)}\right]^T = \mathbf{f}_{\boldsymbol{\Theta}_m}(\mathbf{x}_i)$. Each of the $M$ individual distributions are then combined into a single prediction.

Full heteroscedasticity from unrestricted predictive variance is critical for DEs to produce calibrated output distributions. The total variance of an ensemble prediction can be described as $\text{Var}[y_i|\mathbf{x}_i] = \mathbb{E}_m[\sigma_m^{2\,(i)}] + \text{Var}_m[\mu_m^{(i)}]$, which is equal to the average aleatoric uncertainty of the members, plus the variance of the predicted means (Appendix C.6). If each member of the ensemble is not fully het-

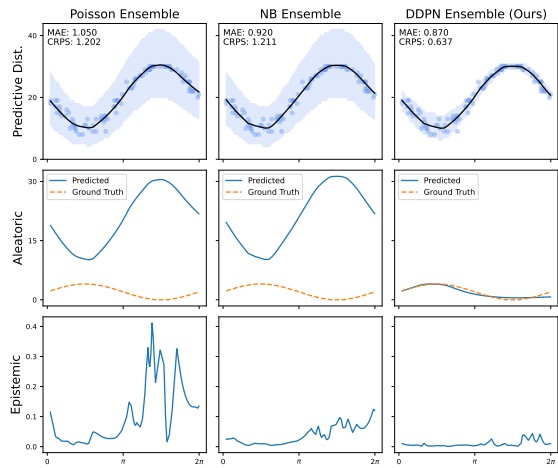

Figure 1: Simulation experiment demonstrating a heteroscedastic data-generating process with discrete outputs. The predictive distributions of three ensemble methods are compared. The mean predictions are shown in black, the 95% credible intervals shaded in blue, along with the corresponding Mean Absolute Error (MAE) and Continuous Ranked Probability Score (CRPS) in the top left (**Top**). Uncertainty is decomposed into its aleatoric (**Middle**) and epistemic (**Bottom**) components, with the ground-truth aleatoric uncertainty represented by a dashed line. Existing discrete regression methods are unable to accurately capture aleatoric uncertainty, which impacts both epistemic and total predictive uncertainty. In contrast, ensembles comprising Deep Double Poisson Networks (DDPNs) effectively represent epistemic and aleatoric uncertainty, while also improving mean fit.

eroscedastic (i.e., the regressor produces constrained variance predictions) the aleatoric term can be misspecified, which induces a miscalibrated predictive distribution of the ensemble. Additionally, recent work has shown that aleatoric and epistemic uncertainty are largely entangled in practice (Mucsányi et al., 2024), further compounding miscalibration. Consequently, poor aleatoric uncertainty quantification will likely also give rise to poor estimates of epistemic uncertainty. For a concrete example, see Figure 1. In this experiment we train three count regression DEs using a synthetic dataset and plot their predictive distributions, along with the estimated aleatoric and epistemic uncertainty decompositions. Two of these DEs, Poisson and Negative Binomial, are constituted of members that are not fully heteroscedastic (Proposition 2.3). This leads to overestimated aleatoric uncertainty and miscalibrated predictive distributions. We also note that such misspecification impacts epistemic uncertainty, which is high in these ensembles despite adequate training data.

While full heteroscedasticity in deep regression on continu-

ous outputs is well-studied (Nix & Weigend, 1994; Bishop, 1994), no such method exists for count data. Previous work trains a network to predict the $\lambda$ parameter of a Poisson distribution and minimize its negative log likelihood (NLL) (Fallah et al., 2009). However, the Poisson parameterization of the neural network suffers from the *equi-dispersion* restriction: the predictive mean and variance are the same ($\hat{\lambda} = \hat{\mu} = \hat{\sigma}^2$). Another common alternative is to train the network to minimize Negative Binomial (NB) NLL (Xie, 2022). The Negative Binomial breaks *equi-dispersion* by introducing another parameter to the PMF. This helps disentangle the mean and variance, but suffers from the *over-dispersion* restriction: $\hat{\sigma}^2 \geq \hat{\mu}$. Neither of these methods can produce unrestricted heteroscedastic variance.

A plausible alternative to introduce full heteroscedasticity to the count regression setting is to violate distributional assumptions and simply apply models with Gaussian likelihoods. However, this decision neutralizes a crucial inductive bias, since it outputs a probability *density* function, $p(y|\mathbf{f}_\Theta(\mathbf{x}))$, $y \in \mathbb{R}$ over a *discrete* output space, i.e. $y \in \mathbb{Z}_{\geq 0}$. This can impact both the accuracy and calibration of the predictive distribution, since the model is constrained to assign probability to infeasible real values and thus cannot maximally concentrate or center its predictions around the ground truth. Such a model will also exhibit various pathologies: 1) it will assign nontrivial probability to negative values (impossible for counting problems) when the predicted mean is small, and 2) the boundaries of its predictive intervals (i.e. highest density interval or 95% credible interval) are likely to fall between two valid integers, diminishing their interpretability and utility.

**Our Contributions**    To address these issues, we introduce the Deep Double Poisson Network (DDPN), a novel discrete neural regression model (see Figure 2). DDPN models aleatoric uncertainty by outputting the parameters of the Double Poisson distribution (Efron, 1986), a highly flexible predictive distribution over $\mathbb{Z}_{\geq 0}$. Unlike existing count regression methods based on the Poisson and Negative Binomial distributions, DDPN exhibits full heteroscedasticity, allowing it to model a broader range of uncertainty patterns. At the same time, DDPN avoids the misspecification issues of Gaussian-based models on count data, enabling sharper, more reliable predictions. In addition to these desirable characteristics, we demonstrate that DDPN possesses robust regression properties akin to those of heteroscedastic Gaussian models. To do so, we propose a formal definition of learnable loss attenuation, a concept first studied by Kendall & Gal (2017) in which a model can adaptively modify its loss function to lower the impact of outlier points during training, and prove that DDPN satisfies this definition. Furthermore, we introduce a discrete analog of the $\beta$-modification proposed by Seitzer et al. (2022), en-

abling controllable attenuation strength. Across a variety of datasets, our experiments show that DDPN (both individually and as an ensemble method) outperforms all baselines in terms of mean accuracy, calibration, and out-of-distribution detection, establishing a new state-of-the-art for deep count regression.

## 2. Modeling Predictive Uncertainty with Neural Networks

A large body of work has developed methods to represent both epistemic and aleatoric uncertainty in deep learning.

### 2.1. Epistemic Uncertainty

Epistemic uncertainty refers to uncertainty due to model misspecification. Modern neural networks tend to be significantly underspecified by the data, which introduces a high degree of uncertainty (Wilson & Izmailov, 2020). A variety of techniques have been proposed to explicitly represent epistemic uncertainty including Bayesian inference (Chen et al., 2014; Hoffman et al., 2014; Wilson & Izmailov, 2020), variational inference (Graves, 2011; Blundell et al., 2015), Laplace approximation (Daxberger et al., 2021), and Epistemic Neural Networks (Osband et al., 2023). Recently, deep ensembles have emerged as a simple and popular solution (Lakshminarayanan et al., 2017; D'Angelo & Fortuin, 2021; Dwaracherla et al., 2022). Other work connects Bayesian inference and ensembles by arguing the latter can be viewed as a Bayesian model average where the posterior is sampled at multiple local modes (Fort et al., 2019; Wilson & Izmailov, 2020). This approach has a number of attractive properties: 1) it generally improves predictive performance (Dietterich, 2000); 2) it can model more complex predictive distributions; and 3) it effectively represents uncertainty over learned weights, which leads to better calibration.

### 2.2. Aleatoric Uncertainty

Aleatoric uncertainty quantifies observation noise and generally cannot be reduced with more data (Der Kiureghian & Ditlevsen, 2009; Kendall & Gal, 2017). In practice, this uncertainty can be introduced by low resolution sensors, blurry images, or the intrinsic noise of a signal.

#### 2.2.1. HETEROSCEDASTIC REGRESSION IN DEEP LEARNING

Aleatoric noise is commonly modeled in deep learning by learning the parameters of a probability distribution over the label. This often takes the form of heteroscedastic regression, where the network learns an input-dependent dispersion parameter, $\hat{\phi}_i$, in addition to an estimate of the mean, $\hat{\mu}_i$ (Nix & Weigend, 1994; Bishop, 1994). In the Gaussian case, $\hat{\phi}_i = \hat{\sigma}_i^2$. Recent work identifies issues with training such

networks via Gaussian NLL due to the the influence of $\hat{\sigma}^2$ on the gradient of the mean, $\hat{\mu}$. Immer et al. (2024) reparameterize their model to output the natural parameters of the Gaussian distribution. Seitzer et al. (2022) propose a modified objective and introduce a hyperparameter, $\beta \in [0, 1]$, which tempers the impact of $\hat{\sigma}^2$ on the gradient of $\hat{\mu}$ and offers tunable control over the loss attenuation properties of the Gaussian NLL. Stirn et al. (2023) modify their architecture to include separate sub-networks for $\hat{\mu}$ and $\hat{\sigma}^2$, then use a stop gradient operation to neutralize the impact of $\hat{\sigma}^2$ on the $\hat{\mu}(\mathbf{x})$ sub-network. In the count setting, Poisson (Fallah et al., 2009) and Negative Binomial (Xie, 2022) likelihoods have been used to train heteroscedastic neural networks.

#### 2.2.2. HETEROSCEDASTIC REGRESSION WITH GENERALIZED LINEAR MODELS

Historically, Generalized Linear Models (GLMs) have been used to model predictive uncertainty on tabular data. GLMs specify a conditional distribution, $p(y_i|\eta_i, \phi)$, where $p$ is a member of the exponential family, $\eta_i = \boldsymbol{w}^T \boldsymbol{x}_i$ represents the natural parameter of $p$, and $\phi$ is the dispersion term (McCullagh, 1989; Murphy, 2023). A link function, $l(\cdot)$, is selected to specify a mapping between the natural parameter and the mean such that $l(\mu_i) = \eta_i = \boldsymbol{w}^T \boldsymbol{x}_i$. The model is then fit by minimizing NLL. Many common models can be viewed under this general framework, including logistic regression, Poisson regression, and binomial regression (Fahrmeir et al., 2013). The Double Poisson distribution we employ in this work was originally developed in the context of GLMs and proposed a constrained dispersion term with an explicit dependence on the mean (Efron, 1986). More recent work employs Joint GLMs with separate covariates for mean and dispersion, allowing for more degrees of freedom (Aragon et al., 2018). All of these models are restricted to linear families of functions.

#### 2.2.3. FULL HETEROSCEDASTICITY

Because misspecification of aleatoric noise often corrupts overall estimates of uncertainty, regression models should be unconstrained in the range of variances they can output. We formalize this property as *full heteroscedasticity*, which ensures that a model's predictive variance is both input-dependent and unconstrained in scale.

**Definition 2.1.** A family of distributions $Q$, parametrized by $\boldsymbol{\psi} \in \mathbb{R}^d$, is said to have *unrestricted variance* if, for any random variable $Z \sim Q$, if we condition on $\mathbb{E}[Z] = \mu$ for any valid $\mu$, for any $\sigma^2 \in (0, \infty)$ there exists a setting of $\boldsymbol{\psi}$ such that $\text{Var}[Z] = \sigma^2$.

**Definition 2.2.** A probabilistic regression model $f$ is called *fully heteroscedastic* if its predictive variance depends on the input $\mathbf{x}_i$ and its output distribution family $Q$ has unrestricted variance.

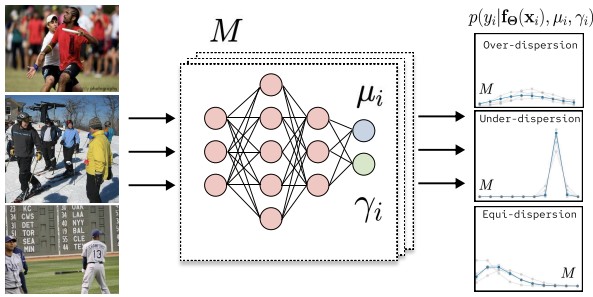

$p(y_i|\mathbf{f}_\Theta(\mathbf{x}_i), \mu_i, \gamma_i)$

Figure 2: An overview of the Deep Double Poisson Network (DDPN). DDPN is a neural network that can process complex data and outputs the parameters of a Double Poisson distribution, $\hat{\mu}_i$ and $\hat{\gamma}_i$. The resulting predictive distributions exhibit unrestricted variance such that the network can learn over-, equi-, and under-dispersion. We ensemble a set of $M$ DDPNs to estimate aleatoric and epistemic uncertainty.

These definitions allow us to characterize existing methods through the lens of full heteroscedasticity.

**Proposition 2.3.** *Gaussian regressors are fully heteroscedastic, whereas Poisson and Negative Binomial regressors are not.*

See Appendix C.1 for a proof. While Gaussian regressors provide the flexibility needed for uncertainty quantification in the continuous setting, existing methods for neural count regression place restrictions on their variance and thus lack full heteroscedasticity. This limitation reduces these models' ability to capture the complex aleatoric noise present in real-world count data.

## 3. Deep Double Poisson Networks (DDPN)

To address the limitations described in the previous section, we propose the Deep Double Poisson Network (DDPN), a state-of-the-art deep regression model that represents families of non-linear functions over complex, non-negative count data. DDPN outputs the parameters of the Double Poisson distribution (Efron, 1986), which results in a fully heteroscedastic predictive distribution under approximate moment assumptions (Proposition 3.1). We demonstrate that DDPN inherits robust regression properties through learnable loss attenuation, similar to Gaussian-based networks. Additionally, we introduce a discrete analog of the $\beta$ modification from Seitzer et al. (2022), allowing tunable control over self-attenuation to improve overall model fit.

We assume access to a dataset, $\mathcal{D}$, with $N$ training examples $\{\mathbf{x}_i, y_i\}_{i=1}^N$, where each $y_i \in \mathbb{Z}_{\geq 0}$ is drawn from some unknown count distribution $p(y_i|\mathbf{x}_i)$. Let $\mathcal{X}$ denote the space of all possible inputs $\mathbf{x}$, let $\mathcal{P}$ denote the space of all possible distributions over $\mathbb{Z}_{\geq 0}$, and let $\psi \in \mathbb{R}^d$ denote a vector of parameters identifying a specific $p \in \mathcal{P}$. We wish to model $\mathcal{P}$ with a neural network $\mathbf{f}_\Theta : \mathcal{X} \to \mathcal{P}$ with

learnable weights $\Theta$ (stacked into $L$ layers). In practice, we model $\mathbf{f}_\Theta : \mathcal{X} \to \psi$. Given such a network, we obtain a predictive distribution, $\hat{p}(y_i|\mathbf{f}_\Theta(\mathbf{x}_i))$, for any input $\mathbf{x}_i$.

In particular, suppose that we restrict our output space to $\mathcal{P}_{DP} \subset \mathcal{P}$, the family of Double Poisson distributions over $y$. Any distribution $p \in \mathcal{P}_{DP}$ is uniquely parameterized by $\psi = [\mu, \gamma]^T \succ \mathbf{0}$, with mean $\mu$ and inverse dispersion $\gamma = \frac{1}{\phi}$. The distribution function, $p : \mathbb{Z}_{\geq 0} \to [0, 1]$, is defined as:

$$p(y|\mu, \gamma) = \frac{\gamma^{\frac{1}{2}} e^{-\gamma\mu}}{c(\mu, \gamma)} \left(\frac{e^{-y} y^y}{y!}\right) \left(\frac{e\mu}{y}\right)^{\gamma y} \quad (1)$$

where $c$ is a normalizing constant. Let $Z$ denote a random variable with a Double Poisson distribution function, then we say $Z \sim \mathrm{DP}(\mu, \gamma)$. In line with previous work (Zou et al., 2013; Aragon et al., 2018), we assume the moment approximations proposed by Efron (1986): $\mathbb{E}[Z] = \mu$ and $\mathrm{Var}[Z] = \frac{\mu}{\gamma}$. We specify a model, [1] $[\log \hat{\mu}_i, \log \hat{\gamma}_i]^T = \mathbf{f}_\Theta(\mathbf{x}_i)$, as follows: let $\mathbf{z}_i = \mathbf{f}_{\Theta_{1:L-1}}(\mathbf{x}_i)$, be the $d$-dimensional hidden representation of the input $\mathbf{x}_i$ produced by the first $L - 1$ layers of a neural network. We apply two separate affine transformations to this representation to obtain our distribution parameters: $\log(\hat{\mu}_i) = \boldsymbol{w}_\mu^T \mathbf{z}_i + b_\mu$ and $\log(\hat{\gamma}_i) = \boldsymbol{w}_\gamma^T \mathbf{z}_i + b_\gamma$.

The flexibility induced by the $\gamma$ parameter in DDPN yields a powerful model with the capacity to represent any form of aleatoric noise. Referring back to Definition 2.2, we concretely state this property in the following proposition (with a proof in Appendix C.2):

**Proposition 3.1.** *DDPN regressors are fully heteroscedastic under approximate moment assumptions.*

See Appendix A.3 for a thorough assessment of the quality of Efron's moment approximations. In practice, we observe that these estimates are almost exact for nearly all relevant values of $\mu$ and $\sigma^2$ (Figure 7).

### 3.1. DDPN Objective

To learn the weights of a DDPN, we minimize the following NLL objective (averaged across all prediction / target tuples $(\hat{\mu}_i, \hat{\gamma}_i, y_i)$ in the dataset):

$$\mathscr{L}_i = \left(-\frac{\log \hat{\gamma}_i}{2} + \hat{\gamma}_i \hat{\mu}_i - \hat{\gamma}_i y_i (1 + \log \hat{\mu}_i - \log y_i)\right) \quad (2)$$

In accordance with convention, we define $y_i \log y_i = 0$ when $y_i = 0$ (Cover, 1999). During training, we mini-

---

[1] For both $\hat{\mu}$ and $\hat{\gamma}$ we apply the log "link" function to ensure positivity and numerical stability. We simply exponentiate whenever $\hat{\mu}_i$ or $\hat{\gamma}_i$ are needed (i.e., to evaluate the P.M.F.)

mize $\mathbb{E}[\mathscr{L}_i]$ iteratively via stochastic gradient descent (or common variants). Note that this objective drops the normalizing constant, $c(\mu_i, \gamma_i)$. Prior work demonstrates that setting $c(\mu_i, \gamma_i) = 1$ provides an accurate approximation of the Double Poisson density and is easier to optimize (Efron, 1986; Chow & Steenhard, 2009). We provide a full derivation of Equation 2 from the Double Poisson distribution function in Appendix A.1.

## 3.2. Loss Attenuation Dynamics of DDPN

Previous work (Kendall & Gal, 2017) states that Gaussian heteroscedastic regressors exhibit *learnable loss attenuation*, a property where predictive uncertainty in the NLL objective reduces the influence of outliers. We will demonstrate that the DDPN objective has this same property. To support this claim, we first introduce a formal definition of learnable attenuation — a concept that, to the best of our knowledge, has not been rigorously defined in prior literature:

**Definition 3.2.** The loss function of a regressor $\mathscr{L}(\hat{\mu}_i, \hat{\phi}_i)$, taking as input the predicted mean $\hat{\mu}_i \in \mathcal{U}$ and dispersion $\hat{\phi}_i \in (0, \infty)$, exhibits *learnable attenuation* if it admits the form $\mathscr{L}(\hat{\mu}_i, \hat{\phi}_i) = d(\hat{\phi}_i) + a(\hat{\phi}_i)r(\hat{\mu}_i, y_i)$, where $\lim_{\hat{\phi} \to \infty} d(\hat{\phi}) = \infty$, $\lim_{\hat{\phi} \to \infty} a(\hat{\phi}) = 0$, and $r \geq 0$ with equality holding iff $\hat{\mu}_i = y_i$. We call $d$ the *dispersion penalty*, $a$ the *attenuation factor*, and $r$ the *residual penalty*.

A key consequence of this formulation is that a model trained with such a loss can actively *dampen* the impact of certain data points—a phenomenon captured in the following proposition:

**Proposition 3.3.** *If a loss function $\mathscr{L}(\hat{\mu}_i, \hat{\phi}_i)$ exhibits learnable attenuation, then $\frac{a(\hat{\phi}_i)r(\hat{\mu}_i, y_i)}{d(\hat{\phi}_i)} \to 0$ as $\hat{\phi}_i \to \infty$.*

This result (proved in Appendix C.3) reveals a fundamental mechanism of learnable attenuation: by increasing the predicted dispersion $\hat{\phi}_i$, a model can effectively nullify the residual penalty's contribution to the total loss. This allows it to "ignore" high-error outliers, leading to a robust regressor with better overall accuracy. Meanwhile, the dispersion penalty in the loss discourages the model from universally inflating $\hat{\phi}_i$, ensuring that it does not suppress all errors.

The Gaussian NLL commonly used to train heteroscedastic regressors naturally satisfies Definition 3.2, as first observed by Kendall & Gal (2017). Specifically, if we set $\hat{\phi}_i = \hat{\sigma}_i^2$, we obtain $d(\hat{\phi}_i) = \frac{1}{2} \log \hat{\phi}_i$, $r(\hat{\mu}_i, y_i) = (\hat{\mu}_i - y_i)^2$, and $a(\hat{\phi}_i) = \frac{1}{2\phi}$.

**Proposition 3.4.** *The DDPN objective in Equation 2 exhibits learnable attenuation of the form $d(\hat{\phi}_i) = \frac{1}{2} \log \hat{\phi}_i$, $a(\hat{\phi}_i) = \frac{1}{\hat{\phi}_i}$, and $r(\hat{\mu}_i, y_i) = (\hat{\mu}_i - y_i) - y_i(\log \hat{\mu}_i - \log y_i)$.*

A proof is provided in Appendix C.4. Because DDPN satisfies the learnable attenuation property, it — like Gaussian

models — can dynamically adjust to downweight the influence of outliers and noisy labels. This robustness likely contributes to the superior performance of DDPN observed in Section 4.

## 3.3. $\beta$-DDPN: Controllable Loss Attenuation

Seitzer et al. (2022) argue that the "learned loss attenuation" property of neural networks trained via Gaussian NLL can sometimes result in premature convergence due to inflated predicted variance in hard-to-fit regions of the training data, which in turn can give rise to suboptimal mean fit. The mechanism that drives this behavior is the presence of the predicted variance term in the partial derivative of the NLL with respect to the mean. We observe that these same phenomena exist with DDPN. Our loss has the following partial derivatives: $\frac{\partial \mathscr{L}_i}{\partial \hat{\mu}_i} = \hat{\gamma}_i \left(1 - \frac{y_i}{\hat{\mu}_i}\right)$ and $\frac{\partial \mathscr{L}_i}{\partial \hat{\gamma}_i} = -\frac{1}{2\hat{\gamma}_i} + \hat{\mu}_i - y_i(1 + \log \hat{\mu}_i - \log y_i)$.

Notice that if the predicted inverse dispersion, $\hat{\gamma}_i$, is sufficiently small (corresponding to large variance), it can completely zero out $\frac{\partial \mathscr{L}_i}{\partial \hat{\mu}_i}$ regardless of the current value of $\hat{\mu}_i$. Thus, during training, a neural network can converge to (and get "stuck" in) suboptimal solutions wherein poor mean fit is explained away via large uncertainty values. To remedy this behavior, we propose a modified loss function, $\mathscr{L}_i^{(\beta)}$, also called the Double Poisson $\beta$-NLL:

$$\mathscr{L}_i^{(\beta)} = \left\lfloor \hat{\gamma}_i^{-\beta} \right\rfloor \left( -\frac{\log \hat{\gamma}_i}{2} + \hat{\gamma}_i \hat{\mu}_i - \hat{\gamma}_i y_i (1 + \log \hat{\mu}_i - \log y_i) \right) \tag{3}$$

where $\lfloor \cdot \rfloor$ denotes the *stop-gradient* operation and we once again average across all prediction / target tuples $(\hat{\mu}_i, \hat{\gamma}_i, y_i)$ in the dataset. With this modification we can effectively temper the loss attenuation behavior of DDPN. We now have partial derivatives $\frac{\partial \mathscr{L}_i^{(\beta)}}{\partial \hat{\mu}_i} = \left(\hat{\gamma}_i^{1-\beta}\right)\left(1 - \frac{y_i}{\hat{\mu}_i}\right)$ and $\frac{\partial \mathscr{L}_i^{(\beta)}}{\partial \hat{\gamma}_i} = -\frac{1}{2\hat{\gamma}_i^{1+\beta}} + \hat{\mu}_i - y_i(1 + \log \hat{\mu}_i - \log y_i)$. The Double Poisson $\beta$-NLL is parameterized by $\beta \in [0, 1]$, where $\beta = 0$ recovers the original Double Poisson NLL and $\beta = 1$ corresponds to fitting the mean, $\mu$, with no respect to $\gamma$ (while still performing normal weight updates to fit the value of $\gamma$). Thus, we can consider the value of $\beta$ as providing a smooth interpolation between the natural DDPN likelihood and a more mean-focused loss. For an empirical demonstration of the impact of $\beta$ on DDPN, see Figure 6 in Section 4.5.

## 3.4. DDPN Ensembles

The formulation in the previous sections describes a network with a single forward pass. As noted in Section 2.1, multiple independently-trained neural networks can be combined to improve mean fit and distributional calibration.

We create DDPN ensembles by combining the predictive distributions of $M$ distinct DDPNs in a uniform mixture as follows: Given $M$ settings of weights $\{\mathbf{\Theta}_m\}_{m=1}^{M}$, we model $y_i$ as $p(y_i|\mathbf{x}_i) = \frac{1}{M}\sum_{m=1}^{M} p(y_i|\mathbf{f}_{\mathbf{\Theta_m}}(\mathbf{x}_i))$. Section 4 convincingly demonstrates that combining model predictions in this way yields the best overall performance.

# 4. Experiments

In our experiments, we aim to answer the following research questions: (1) Is DDPN flexible enough to fit any count distribution, even under known misspecification? (2) How does DDPN compare to existing deep regression methods as measured by accuracy and calibration? (3) Does the enhanced uncertainty quantification of DDPN from full heteroscedasticity combined with proper inductive biases of a discrete predictive distribution lead to better out-of-distribution detection? (4) What is the effect of the $\beta$ hyperparameter on the training dynamics of DDPN?

## 4.1. Evaluation Metrics

We evaluate each regression method along two key dimensions: *accuracy*, measured by mean fit, and *calibration*, which indicates the overall quality of the predictive distribution. Accuracy is quantified using the Mean Absolute Error (MAE). Calibration is assessed using the Continuous Ranked Probability Score (CRPS), a strictly proper scoring rule (Matheson & Winkler, 1976), with lower values signifying closer alignment between the predictive CDF $F_i$ and the observed label $y_i$. Detailed explanations and definitions of these evaluation metrics are provided in Appendix B.2.

## 4.2. Misspecification Recovery

DDPN is trained by minimizing Double Poisson NLL, which makes a distributional assumption about the targets. In this section, we test whether DDPN is robust to violations of this assumption. To do so, we generate data from two processes: 1) $Y|X \sim \text{Poisson}(\exp(\frac{1}{2}X))$; and 2) $Y|X \sim \text{NegBinom}(X^2, \frac{1}{2})$; neither of which conform to the assumptions in Section 3. For each dataset, we train both a DDPN and a model explicitly matched to the noise distribution of the data (a Poisson DNN for the first process and a Negative Binomial DNN for the second). Figures 3a and 3b compare the predictive distributions learned by these models. Within each figure, we report MAE and CRPS.

Remarkably, DDPN is flexible enough to accurately capture the aleatoric structure of the data in both experiments. It achieves performance that meets or exceeds models explicitly tailored to the true noise distribution. These results highlight the robustness of DDPN and suggest that it offers a reliable and versatile approach to neural count regression, even in the face of potential misspecification.

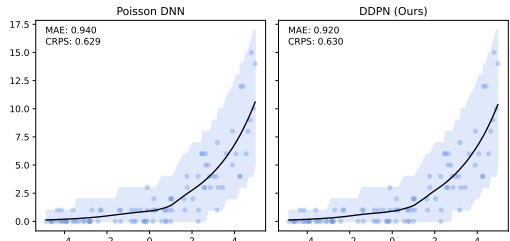

(a) Predictive distributions of a Poisson DNN and DDPN on data where $Y|X \sim$ Poisson.

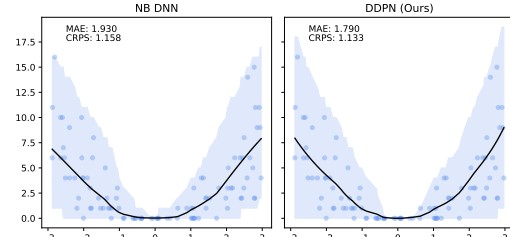

(b) Predictive distributions of a Negative Binomial (NB) DNN and DDPN on data where $Y|X \sim$ NegBinom.

Figure 3: Even when explicitly misspecified, DDPN recovers the data-generating distribution.

| Dataset | Modality | Backbone |
|---|---|---|
| Length of Stay | Tabular | MLP |
| COCO-People | Image | ViT-B-16 |
| Inventory | Point Cloud | CountNet3D |
| Reviews | Text | DistilBERT Base Cased |

Table 1: Summary of datasets used, along with their corresponding backbones.

## 4.3. Real-World Datasets

We compare DDPN to baselines on a number of real-world, complex count datasets. A summary of the datasets used and the corresponding backbone architectures is given in Table 1. Notably, we run experiments across various modalities, including tabular, image, point cloud, and text data.

Length of Stay (Microsoft, 2016) is a tabular dataset where the task is to forecast the number of days of a patient will spend in a hospital. The features consist of patient measurements and attributes of the healthcare facility. COCO-People (Lin et al., 2014) is an adaptation of the MS-COCO dataset, where the task is to predict the number of people in each image. Inventory (Jenkins et al., 2023) consists of point clouds that capture retail shelves. The goal is to predict the number of products in the point cloud. Finally, Reviews is taken from the "Patio, Lawn, and Garden" category of the Amazon Reviews dataset (Ni et al., 2019) . The goal is to predict the discrete 1-5 product rating from the text review.

Table 2: Results across four datasets of differing modalities: Length of Stay (tabular), COCO-People (image), Inventory (point cloud), and Amazon Reviews (language). We denote the best performer in each subgroup (aleatoric only vs. aleatoric + epistemic) in **bold** and the second-best with an underline. In each case, DDPN outperforms all baselines in terms of MAE and CRPS, achieving a new state-of-the-art on count regression tasks.

| | | Length of Stay | | COCO-People | | Inventory | | Reviews | |
|---|---|---|---|---|---|---|---|---|---|
| | | MAE ($\downarrow$) | CRPS ($\downarrow$) | MAE ($\downarrow$) | CRPS ($\downarrow$) | MAE ($\downarrow$) | CRPS ($\downarrow$) | MAE ($\downarrow$) | CRPS ($\downarrow$) |
| Aleatoric Only | Poisson DNN | 0.664 (0.01) | 0.553 (0.01) | 1.099 (0.02) | 0.851 (0.01) | 1.023 (0.04) | 0.706 (0.01) | 0.818 (0.01) | 0.559 (0.00) |
| | NB DNN | 0.685 (0.00) | 0.570 (0.00) | 1.143 (0.05) | 0.867 (0.01) | 1.020 (0.04) | 0.708 (0.01) | 0.855 (0.01) | 0.562 (0.00) |
| | Gaussian DNN | 0.599 (0.01) | 0.453 (0.02) | 1.219 (0.12) | 0.866 (0.07) | 0.936 (0.01) | 0.659 (0.00) | 0.452 (0.01) | 0.323 (0.00) |
| | Faithful Gaussian | 0.582 (0.00) | 0.436 (0.01) | 1.082 (0.01) | 0.879 (0.01) | 0.959 (0.03) | 0.688 (0.02) | 0.428 (0.00) | 0.428 (0.00) |
| | Natural Gaussian | 0.597 (0.01) | 0.439 (0.01) | 1.157 (0.04) | 0.848 (0.02) | 0.958 (0.01) | 0.675 (0.00) | 0.428 (0.00) | 0.312 (0.00) |
| | $\beta_{0.5}$-Gaussian | 0.600 (0.01) | 0.427 (0.01) | 1.055 (0.01) | 0.786 (0.00) | 0.935 (0.01) | 0.669 (0.01) | 0.420 (0.00) | 0.306 (0.00) |
| | $\beta_{1.0}$-Gaussian | 0.646 (0.01) | 0.462 (0.01) | 1.085 (0.01) | 0.809 (0.00) | 0.923 (0.01) | 0.653 (0.01) | 0.458 (0.01) | 0.327 (0.00) |
| | DDPN (ours) | **0.502** (0.01) | 0.390 (0.04) | 1.135 (0.08) | 0.810 (0.03) | 0.906 (0.01) | **0.632** (0.01) | 0.392 (0.01) | 0.277 (0.00) |
| | $\beta_{0.5}$-DDPN (ours) | 0.516 (0.01) | **0.370** (0.01) | 1.095 (0.03) | 0.782 (0.02) | **0.905** (0.02) | 0.635 (0.01) | **0.356** (0.01) | 0.268 (0.00) |
| | $\beta_{1.0}$-DDPN (ours) | 0.558 (0.01) | 0.407 (0.01) | **1.006** (0.01) | **0.759** (0.01) | 0.909 (0.01) | 0.634 (0.01) | **0.356** (0.00) | **0.263** (0.00) |
| Aleatoric + Epistemic (DEs) | Poisson DNN | 0.650 | 0.547 | 1.046 | 0.817 | 0.996 | 0.683 | 0.823 | 0.556 |
| | NB DNN | 0.681 | 0.567 | 1.066 | 0.824 | 0.982 | 0.686 | 0.857 | 0.560 |
| | Gaussian DNN | 0.590 | 0.450 | 1.148 | 0.815 | 0.902 | 0.634 | 0.447 | 0.319 |
| | Faithful Gaussian | 0.571 | 0.429 | 1.042 | 0.841 | 0.909 | 0.643 | 0.424 | 0.324 |
| | Natural Gaussian | 0.582 | 0.428 | 1.090 | 0.800 | 0.916 | 0.643 | 0.423 | 0.307 |
| | $\beta_{0.5}$-Gaussian | 0.591 | 0.420 | 1.019 | 0.740 | 0.879 | 0.619 | 0.414 | 0.302 |
| | $\beta_{1.0}$-Gaussian | 0.633 | 0.453 | 1.050 | 0.765 | 0.887 | 0.624 | 0.455 | 0.324 |
| | DDPN (ours) | **0.485** | 0.361 | 1.024 | 0.744 | 0.861 | 0.604 | 0.373 | 0.268 |
| | $\beta_{0.5}$-DDPN (ours) | 0.495 | **0.359** | 1.029 | 0.729 | **0.840** | **0.590** | 0.358 | 0.261 |
| | $\beta_{1.0}$-DDPN (ours) | 0.543 | 0.393 | **0.959** | **0.712** | 0.859 | 0.597 | **0.344** | **0.257** |

#### 4.3.1. BASELINES

We compare to two discrete baselines: 1) Poisson DNN (Fallah et al., 2009) and 2) Negative Binomial DNN (Xie, 2022). We also include fully heteroscedastic continuous models in our experiments: 1) Gaussian DNN (Nix & Weigend, 1994), 2) $\beta$-Gaussian (Seitzer et al., 2022), 3) Faithful Gaussian (Stirn et al., 2023), and 4) Natural Gaussian (Immer et al., 2024). For $\beta$-Gaussian we use the prescribed values of $\beta = 0.5$ and $\beta = 1.0$ from Seitzer et al. (2022), and mirror this behavior for our discrete adaptation, $\beta$-DDPN. A subscript marks the specific setting of $\beta$ (e.g. $\beta_{0.5}$).

We also evaluate ensembles for all methods to highlight the effects of modeling both aleatoric and epistemic uncertainty. Gaussian ensembles are produced according to the technique of Lakshminarayanan et al. (2017), while DDPN, Poisson, and Negative Binomial ensembles follow the prediction strategy outlined in Section 3.4. See Table 2 for results.

For each dataset, we generate train/val/test splits with a fixed random seed. We select models according to lowest average loss on the validation split and report results on the test split. We train and evaluate 5 models per technique and record the empirical mean and standard deviation of each metric. To form ensembles, these same 5 models are combined. All experiments are implemented in PyTorch (Paszke et al., 2017). Details related to data splits, network architecture, hardware, and hyperparameter selection are reported in Appendix B.3. Source code is available online[2].

---
[2] https://github.com/delicious-ai/ddpn

#### 4.3.2. ANALYSIS OF EMPIRICAL RESULTS

We observe that DDPN (or one of its $\beta$ variants) achieves the best accuracy and calibration on every task we benchmark. In almost all cases, the margin is substantial— across both individual models and ensembles, COCO-People is the only dataset in which a non-DDPN model ($\beta_{0.5}$-Gaussian) even places in the top three. In line with previous results (Lakshminarayanan et al., 2017; Fort et al., 2019), we find that marginalizing over multiple plausible sets of weights through deep ensembles yields improvements in both mean fit and probabilistic alignment. Ensembled DDPN also outperforms all baselines in terms of accuracy and calibration.

On most datasets, we observe that Negative Binomial and Poisson DNNs struggle with fitting the mean in addition to uncertainty quantification. In contrast to Gaussian and DDPN models that can perform loss attenuation during training (see Section 3.2), Negative Binomial and Poisson regressors are limited in their ability to discount outliers. Whenever these models predict a high count, the equidispersion and overdispersion assumptions built into each respective distribution force a high uncertainty to be predicted as well. Thus, training data points are weighted according to their predicted labels without truly accounting for observation noise, and outliers are not properly tempered.

The gap in performance between Gaussian neural regressors and DDPN models, meanwhile, can largely be attributed to the lack of an inductive bias: Gaussian models naïvely assign probability mass to continuous values that are known a priori to be infeasible, while DDPNs concentrate their

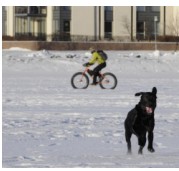 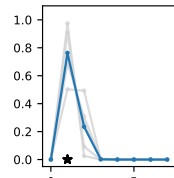 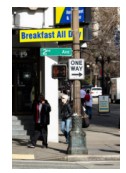 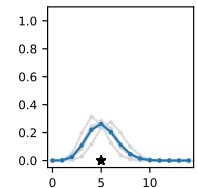 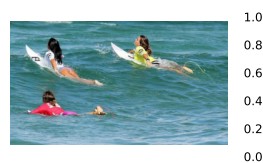 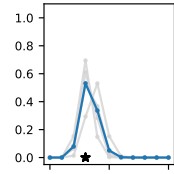

Figure 4: Predictive PMFs from a $\beta_{1.0}$-DDPN ensemble on samples from the test split of `COCO-People`. Individual member predictions are in gray, while the ensemble prediction is in blue. DDPN is able to flexibly and accurately represent counts of various magnitudes.

probabilities on discrete counts by explicit construction. This mismatch produces predictive CDFs that are more misaligned with observations compared to those produced by DDPNs. See Figure 5 for two illustrative examples from `Length of Stay`.

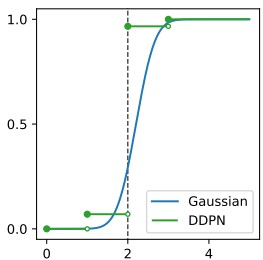 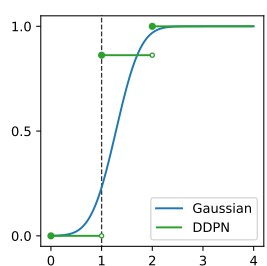

Figure 5: Predictive CDFs from a $\beta_{0.5}$-Gaussian (blue) and $\beta_{0.5}$-DDPN (green) model on two test points from `Length of Stay`. Observed labels are marked with a vertical dashed line. When $y$ is discrete, Gaussian models' predictions suffer from having to assign probability mass to infeasible continuous values. DDPN is free from this constraint.

In Section 3.3, we explain how Seitzer et al. (2022)'s $\beta$ modification can be adapted to improve mean fit in the discrete heteroscedastic setting by defining $\beta$-DDPN. Our results demonstrate that this adjustment to our training objective generally results in superior MAE, although the strength of the effect varies by dataset. We note that CRPS often improves alongside MAE, indicating that $\beta$ can simultaneously boost both accuracy and overall probabilistic fit.

Figure 4 depicts several test instances from `COCO-People` and the predictive distributions produced by a $\beta_{1.0}$-DDPN ensemble, the highest scoring method. Further examples, including predictions from each baseline model, can be found in Appendix D.1.

## 4.4. Out-of-Distribution Detection

In this section, we investigate how DDPN models perform relative to baselines when their predictive distributions are

used for out-of-distribution (OOD) detection. In accordance with common practice, we form scoring threshold-based OOD detectors and evaluate them in the binary classification setting (Hendrycks & Gimpel, 2016; Alemi et al., 2018; Ren et al., 2019). We define our in-distribution (ID) dataset $\mathcal{D}_{in}$ to be `Amazon Reviews` and form our out-of-distribution dataset $\mathcal{D}_{out}$ by compiling verses from the King James Version of the Holy Bible. We refer to the train and test splits of $\mathcal{D}_{in}$ as $\mathcal{D}_{in}^{(trn)}$ and $\mathcal{D}_{in}^{(test)}$ respectively.

We use the total variance of ensemble predictions as our OOD score, since both aleatoric and epistemic uncertainty are helpful OOD indicators (Wang & Aitchison, 2021). The OOD threshold $\tau_\alpha$ is chosen such that we expect a false positive rate of $\alpha$ on data drawn from $\mathcal{D}_{in}$. Specifically, we randomly select 20% of $\mathcal{D}_{in}^{(test)}$ and define $\tau_\alpha$ to be the $(1 - \alpha)$ quantile of the predictive uncertainties on that holdout set. Any inputs from $\mathcal{D}_{in}^{(test)}$ (excluding the held-out data) or $\mathcal{D}_{out}$ that produce a predictive variance above $\tau_\alpha$ are classified as OOD, while the rest are marked as ID.

For each ensemble, we vary $\alpha$ from 0 to 1 and report the resultant area under the ROC curve (AUROC), the area under the precision-recall curve (AUPR), and the false positive rate at the 80% true positive rate (FPR80). To account for variability due to randomness in the calculation of $\tau_\alpha$, we run each evaluation 10 times, re-sampling the holdout set with each iteration, and report the mean and standard deviation of each metric. Results are presented in Table 3.

Our results demonstrate that uncertainties obtained from DDPN (and $\beta$-DDPN) ensembles are the best-suited among all other models evaluated for identifying OOD inputs. This is strong evidence that uncertainty estimates obtained from DDPN models are robust, yielding important context for practitioners around how much an individual prediction can be trusted. For plots showing how distributions differ for ID / OOD data, see Appendix D.2. In particular, Figure 30 highlights the effective OOD behavior of DDPN.

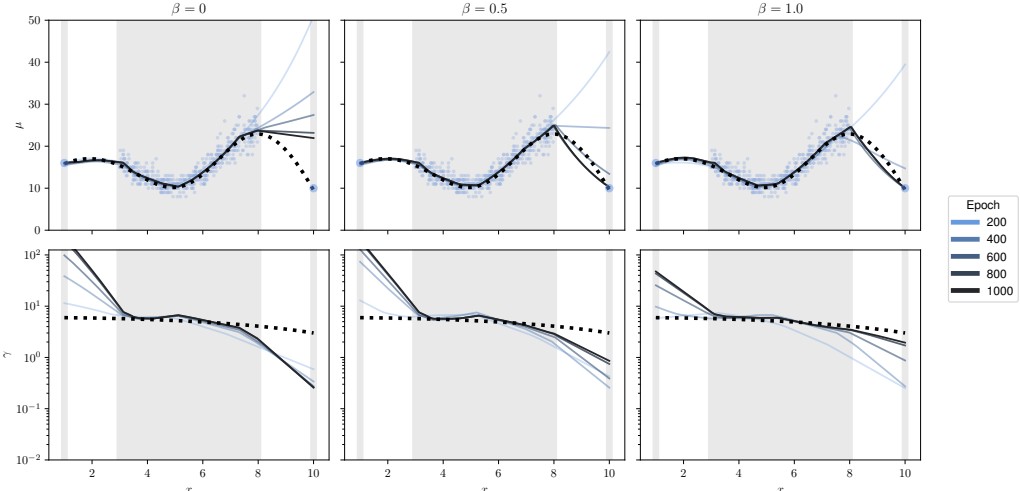

Figure 6: The effect of $\beta$ on the convergence of a DDPN during training, inspired by Fig. 2 of Stirn et al. (2023). Dotted lines indicate ground truth values of $\mu$ and $\gamma$ respectively, while solid lines show the model's learned distribution. Shaded regions illustrate training data coverage. With standard NLL ($\beta = 0$), poor mean fit on the rightmost isolated point is "explained away" via high uncertainty (low values of $\gamma$), leading to subpar convergence. Increasing the value of $\beta$ changes training priorities and allows the network to adequately model the mean without exploding uncertainty estimates. Higher values of $\beta$ lead to faster convergence to the mean; when $\beta = 0.5$, the mean is fit by epoch 800, but when $\beta = 1.0$, the mean is fit by epoch 600.

Table 3: OOD detection results. We denote the best performer for each metric in **bold** and the second-best underlined.

| | AUROC ($\uparrow$) | AUPR ($\uparrow$) | FPR80 ($\downarrow$) |
|---|---|---|---|
| Poisson DNN | 0.330 (0.001) | 0.413 (0.000) | 0.793 (0.001) |
| NB DNN | 0.280 (0.001) | 0.397 (0.000) | 0.819 (0.002) |
| Gaussian DNN | 0.840 (0.001) | 0.812 (0.005) | 0.318 (0.002) |
| Faithful Gaussian | 0.731 (0.001) | 0.670 (0.001) | 0.380 (0.002) |
| Natural Gaussian | 0.836 (0.001) | 0.827 (0.002) | 0.317 (0.002) |
| $\beta_{0.5}$-Gaussian | 0.829 (0.001) | 0.797 (0.004) | 0.323 (0.002) |
| $\beta_{1.0}$-Gaussian | 0.817 (0.001) | 0.806 (0.002) | 0.338 (0.001) |
| DDPN (ours) | 0.854 (0.001) | 0.849 (0.003) | 0.269 (0.002) |
| $\beta_{0.5}$-DDPN (ours) | **0.887** (0.001) | **0.875** (0.003) | **0.199** (0.001) |
| $\beta_{1.0}$-DDPN (ours) | 0.870 (0.001) | 0.851 (0.002) | 0.236 (0.002) |

### 4.5. The Effect of $\beta$ on DDPN Training Dynamics

In this section, we study how the $\beta$ hyperparameter mediates the functional evolution of DDPN's predictive distribution over the course of training. Figure 6 shows a one dimensional count regression problem with three settings of $\beta$. We generate data from $Y|X \sim \text{DP}(\lceil X\sin(X) + 15\rceil, 6 - 0.03X^2)$, where $X \sim \text{Uniform}[3, 8]$. We then produce a set of isolated points at $(1, \lceil\sin(1) + 15\rceil)$ and $(10, \lceil 10\sin(10) + 15\rceil)$ and concatenate them with the generated dataset. Since these points are relatively far from the rest of the training data, a neural network will struggle to initially fit them. If the network is heteroscedastic, it may downweight the contribution of these outlier points to the overall loss by assigning high uncertainty values (low

values of $\gamma$ for DDPN) to explain the poor fit. This is a consequence of the learnable loss attenuation property from Definition 3.2. This raises the possibility that model training converges without producing a function that fits the outlier points. We can temper the effect of such loss attenuation by training with $\beta$-NLL. In Figure 6, we see that increasing the value of $\beta$ indeed addresses this behavior, with higher $\beta$ leading to faster convergence to the true mean fit. This matches the analysis performed by (Seitzer et al., 2022), who use $\beta$ to regularize the Gaussian NLL.

## 5. Conclusion

In this paper, we presented the Deep Double Poisson Network (DDPN) as a novel approach to enhance the quality of predictive distributions in deep count regression. DDPN achieves full heteroscedasticity through the unrestricted variance of its output distribution, allowing it to accurately capture aleatoric uncertainty of any form. This capability results in uncorrupted estimates of epistemic uncertainty when predictions are combined in an ensemble. In addition, we formally defined the concept of learnable loss attenuation and proved that DDPN exhibits this property. We also proposed the $\beta$-DDPN modification, which enables control over the strength of this loss attenuation to further improve performance. Experiments on diverse datasets demonstrate that DDPN surpasses current baselines in accuracy, calibration, and out-of-distribution detection, establishing a new state-of-the-art in deep count regression.

## Acknowledgments

The work was partially supported by NSF awards #2421839, NAIRR #240120. The views and conclusions contained in this paper are those of the authors and should not be interpreted as representing any funding agencies.

## Impact Statement

By enabling robust uncertainty estimation on count data, DDPN improves the reliability and robustness of AI systems in high-stakes applications such as healthcare, finance, and environmental modeling. The ability to model both epistemic and aleatoric uncertainty enhances trustworthiness, particularly in safety-critical systems. However, as with any predictive modeling approach, there are ethical considerations regarding bias, misuse, and overreliance on model outputs. Ensuring that DDPN is used responsibly requires careful dataset curation, bias mitigation, and transparent reporting of model uncertainties.

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

## A. Deep Double Poisson Networks (DDPNs)

### A.1. Derivation of the DDPN Objective

If we model our regression targets as $y_i \sim \text{DP}(\mu_i, \gamma_i)$, where $[\mu_i, \gamma_i]^T = \mathbf{f}_\Theta(\mathbf{x}_i)$, then the Double Poisson NLL for an individual prediction is obtained as follows:

$$
\begin{aligned}
\operatorname*{argmax}_{\Theta} \left[ p(y_i | \mathbf{f}_\Theta(\mathbf{x}_i)) \right] &= \operatorname*{argmax}_{\mu_i, \gamma_i} \left[ p(y_i | \mu_i, \gamma_i) \right] \\
&= \operatorname*{argmin}_{\mu_i, \gamma_i} \left[ -\log p(y_i | \mu_i, \gamma_i) \right] \\
&= \operatorname*{argmin}_{\mu_i, \gamma_i} \left[ -\log \left( \gamma_i^{\frac{1}{2}} \exp\left( -\gamma_i \mu_i \right) \left( \frac{\exp\left( -y_i \right) y_i^{y_i}}{y_i!} \right) \left( \frac{e\mu_i}{y_i} \right)^{\gamma_i y_i} \right) \right] \\
&= \operatorname*{argmin}_{\mu_i, \gamma_i} \left[ -\log \left( \gamma_i^{\frac{1}{2}} \exp\left( -\gamma_i \mu_i \right) \left( \frac{e\mu_i}{y_i} \right)^{\gamma_i y_i} \right) \right] \\
&= \operatorname*{argmin}_{\mu_i, \gamma_i} \left[ -\left( \frac{1}{2} \log \gamma_i - \gamma_i \mu_i + \gamma_i y_i (1 + \log \mu_i - \log y_i) \right) \right]
\end{aligned}
$$

Thus,

$$
\mathscr{L}_i(y_i, \mu_i, \gamma_i) = -\frac{\log \gamma_i}{2} + \gamma_i \mu_i - \gamma_i y_i (1 + \log \mu_i - \log y_i) \tag{4}
$$

During training, we minimize this NLL on average across $N$ data points $(\mathbf{x}_i, y_i)_{i=1}^N$. N

Note that our derivation assumes $c(\mu_i, \gamma_i) = 1$, which is in line with Fact 1 (Equation 2.10) of Efron (1986). See also Chow & Steenhard (2009), who employ a similar assumption in their work with Double Poisson regression.

### A.2. Limitations

DDPNs are general, easy to implement, and can be applied to a variety of datasets. However, some limitations do exist. One limitation that might arise is on count regression problems of very high frequency (i.e., on the order of thousands or millions). In this paper, we don't study the behavior of DDPN relative to existing benchmarks on high counts. In this scenario, it is possible that the choice of a Gaussian as the predictive distribution may offer a good approximation, even though the regression targets are discrete.

Although we follow precedent (Efron, 1986; Zou et al., 2013; Aragon et al., 2018) in employing the general approximations $\mathbb{E}[Z] \approx \mu$ and $\text{Var}[Z] \approx \frac{\mu}{\gamma}$ for some $Z \sim \text{DP}(\mu, \gamma)$ in this work, these are not guaranteed to hold under all conditions. See Appendix A.3 for a detailed study of these approximations.

One difficulty that can sometimes arise when training a DDPN (or any model trained with NLL) is poor convergence of the model weights. In preliminary experiments for this research, we had trouble obtaining consistently high-performing solutions with the SGD (Kiefer & Wolfowitz, 1952) and Adam (Kingma & Ba, 2014) optimizers, thus AdamW (Loshchilov & Hutter, 2017) was used instead. Similar to Seitzer et al. (2022), we found that the $\beta$-NLL exhibited greater stability across a wider variety of hyperparameter settings. Future researchers using the DDPN technique should be wary of this behavior.

In this paper, we performed a single out-of-distribution (OOD) experiment on `Amazon Reviews`. This experiment provided encouraging evidence of the efficacy of DDPN for OOD detection. However, the conclusions drawn from this experiment may be somewhat limited in scope since the experiment was performed on a single dataset and task. Future work should seek to build off of these results to more fully explore the OOD properties of DDPN on other count regression tasks.

### A.3. Assessing the Quality of Efron's Moment Approximations

In line with prior work, we use Efron's approximations for the first two moments of the Double Poisson distribution in the proof of Proposition 3.1. An obvious question that arises is *how reliable are these approximate moment assumptions*. We introduce the concept of moment-deviation functions (MDFs) to assess this theoretically in Definition A.1.

**Definition A.1.** Let $Q$ be a family of univariate distributions parametrized by $\psi \in \mathbb{R}^d$. Suppose we are given $\hat{\psi}_n : \mathbb{R}^n \to \mathbb{R}^d$, which outputs a setting of parameters for $Q$, $\hat{\psi}_n(\boldsymbol{\mu})$, such that the first $n$ moments of $Z \sim Q_{\hat{\psi}_n(\boldsymbol{\mu})}$, $(\langle Z^1 \rangle, ..., \langle Z^n \rangle)$, are nearly equal to target moments $\boldsymbol{\mu} = (\mu_1, ..., \mu_n)$. Then for any pair $(Q, \hat{\psi}_n)$, $\{\varepsilon_i : \mathbb{R}^n \to \mathbb{R}\}_{i=1}^n$ are moment-deviation functions if, for any valid $\boldsymbol{\mu}$, if $Z \sim Q_{\hat{\psi}_n(\boldsymbol{\mu})}$, we have $|\langle Z^i \rangle - \mu_i| \leq \varepsilon_i(\boldsymbol{\mu})$ for all $1 \leq i \leq n$.

We focus on $n = 2$, (approximation error for the mean / variance of a distribution). If we can pick $\hat{\psi}$ so that $\varepsilon_1, \varepsilon_2$ are small, it follows that $Q$ is incredibly flexible, as there exists a setting of parameters that can roughly achieve any mean and variance. In the Gaussian case $(\mathcal{N}, \mathbb{I})$, we have $\varepsilon_1 = \varepsilon_2 = 0$. The Double Poisson case (using Efron's approximations for $\hat{\psi}$), is handled in Proposition A.2.

**Proposition A.2.** *Let $DP$ denote the Double Poisson family. Set $\hat{\psi}_2(\mu_0, \sigma_0^2) = (\mu_0, \frac{\mu_0}{\sigma_0^2})$. Letting $\gamma_0 = \frac{\mu_0}{\sigma_0^2}$, the MDFs for $(DP, \hat{\psi}_2)$ are:*

$$\varepsilon_1(\mu_0, \sigma_0^2) = \left| \frac{\sum_{y=0}^{\infty} s(\mu_0, \gamma_0, y)(y - \mu_0)}{\sum_{y=0}^{\infty} s(\mu_0, \gamma_0, y)} \right|$$

$$\varepsilon_2(\mu_0, \sigma_0^2) = \left| \frac{d(\mu_0, \gamma_0)\gamma_0^{\frac{1}{2}} \sum_{y=0}^{\infty} s(\mu_0, \gamma_0, y) - \gamma_0 (\sum_{y=0}^{\infty} s(\mu_0, \gamma_0, y)(y - \mu_0))^2}{\gamma_0 (\sum_{y=0}^{\infty} s(\mu_0, \gamma_0, y))^2} \right|$$

*where we have*

$$h(z) = \frac{e^{-z} z^z}{z!}$$

$$r(\mu, \gamma, z) = \gamma(z - \mu + z \log \mu - z \log z)$$

$$s(\mu, \gamma, z) = h(z) \exp(r(\mu, \gamma, z))$$

$$d(\mu, \gamma) = \gamma^{-1/2} \left[ \sum_{y=0}^{\infty} s(\mu, \gamma, y)(\gamma(y - \mu)^2 - y) + \sum_{y=0}^{\infty} s(\mu, \gamma, y)(y - \mu) \right].$$

*Proof.* All we have to show is that the mean and variance of $Z \sim Q_{\hat{\psi}_2(\mu_0, \sigma_0^2)}$ are within the provided $\varepsilon_1$ and $\varepsilon_2$.

We first re-cast the Double Poisson PMF into canonical form, with natural parameters $\eta_1 = \gamma \log \mu$ and $\eta_2 = \gamma$:

$$
\begin{aligned}
p(z|\mu, \gamma) &= \frac{\gamma^{\frac{1}{2}} e^{-\gamma\mu}}{c(\mu, \gamma)} \left( \frac{e^{-z} z^z}{z!} \right) \left( \frac{e\mu}{z} \right)^{\gamma z} \\
&= \frac{\gamma^{\frac{1}{2}} e^{-\gamma\mu}}{c(\mu, \gamma)} h(z) \exp\{\gamma z + \gamma z \log \mu - \gamma z \log z)\} \\
&= \frac{\eta_2^{\frac{1}{2}} \exp\{-\eta_2 \exp\{\frac{\eta_1}{\eta_2}\}\}}{c(\exp\{\frac{\eta_1}{\eta_2}\}, \eta_2)} h(z) \exp\{\eta_1 z + \eta_2 z - \eta_2 z \log z\} \\
&= h(z) \exp\{\boldsymbol{\eta}^T \boldsymbol{T}(z) - A(\boldsymbol{\eta})\}
\end{aligned}
$$

where $h(z)$ matches the form specified in the proposition, $\boldsymbol{T}(z) = (z, z(1 - \log z))$ is the sufficient statistic, and $A(\boldsymbol{\eta}) = \eta_2 \exp\{\frac{\eta_1}{\eta_2}\} - \frac{\log \eta_2}{2} + \log c(\exp\{\frac{\eta_1}{\eta_2}\}, \eta_2)$ is the cumulant function.

By properties of the canonical form, we then have $\mathbb{E}[Z] = \frac{\partial A}{\partial \eta_1}$, $\text{Var}[Z] = \frac{\partial^2 A}{\partial \eta_1^2}$. Letting $\mathscr{C}(\eta_1, \eta_2) = c(\exp\{\frac{\eta_1}{\eta_2}\}, \eta_2)$, we obtain

$$\mathbb{E}[Z] = \exp\left\{\frac{\eta_1}{\eta_2}\right\} + \frac{\frac{\partial}{\partial\eta_1}\mathscr{C}(\eta_1,\eta_2)}{\mathscr{C}(\eta_1,\eta_2)}$$

$$\text{Var}[Z] = \frac{\partial}{\partial\eta_1}\left(\exp\left\{\frac{\eta_1}{\eta_2}\right\} + \frac{\frac{\partial}{\partial\eta_1}\mathscr{C}(\eta_1,\eta_2)}{\mathscr{C}(\eta_1,\eta_2)}\right)$$

$$= \frac{\exp\left\{\frac{\eta_1}{\eta_2}\right\}}{\eta_2} + \frac{\mathscr{C}(\eta_1,\eta_2)\frac{\partial^2}{\partial\eta_1^2}\mathscr{C}(\eta_1,\eta_2) - \left(\frac{\partial}{\partial\eta_1}\mathscr{C}(\eta_1,\eta_2)\right)^2}{\mathscr{C}(\eta_1,\eta_2)^2}$$

We first solve for $\mathbb{E}[Z]$. Recall that $c(\mu,\gamma) = \sum_{y=0}^{\infty} \gamma^{\frac{1}{2}} e^{-\gamma\mu} h(y) \exp\{\gamma y + \gamma y \log\mu - \gamma y \log y\}$, or equivalently $c(\mu,\gamma) = \sum_{y=0}^{\infty} \gamma^{\frac{1}{2}} h(y) \exp\{r(\mu,\gamma,y)\}$. Then

$$\frac{\partial}{\partial\eta_1}\mathscr{C}(\eta_1,\eta_2) = \frac{\partial c}{\partial\mu}\frac{\partial\mu}{\partial\eta_1} = \frac{\exp\left\{\frac{\eta_1}{\eta_2}\right\}}{\eta_2}\frac{\partial c}{\partial\mu}$$

$$= \frac{\mu}{\gamma}\sum_{y=0}^{\infty}\gamma^{\frac{1}{2}}h(y)\exp\{r(\mu,\gamma,y)\}\frac{\partial r}{\partial\mu}$$

$$= \sum_{y=0}^{\infty}\gamma^{\frac{1}{2}}h(y)\exp\{r(\mu,\gamma,y)\}(y-\mu)$$

so, recalling that $\exp\left\{\frac{\eta_1}{\eta_2}\right\} = \mu$, we have

$$\mathbb{E}[Z] = \mu + \left(\frac{\sum_{y=0}^{\infty}h(y)\exp\{r(\mu,\gamma,y)\}(y-\mu)}{\sum_{y=0}^{\infty}h(y)\exp\{r(\mu,\gamma,y)\}}\right)$$

$$= \mu + \left(\frac{\sum_{y=0}^{\infty}s(\mu,\gamma,y)(y-\mu)}{\sum_{y=0}^{\infty}s(\mu,\gamma,y)}\right)$$

Thus, if $\hat{\psi}_2 = (\mu_0, \frac{\mu_0}{\sigma_0^2})$, we have

$$\mathbb{E}[Z] - \mu_0 = \left(\frac{\sum_{y=0}^{\infty}s\left(\mu_0,\frac{\mu_0}{\sigma_0^2},y\right)(y-\mu_0)}{\sum_{y=0}^{\infty}s\left(\mu_0,\frac{\mu_0}{\sigma_0^2},y\right)}\right)$$

This implies that

$$|\mathbb{E}[Z] - \mu_0| = \left|\frac{\sum_{y=0}^{\infty}s\left(\mu_0,\frac{\mu_0}{\sigma_0^2},y\right)(y-\mu_0)}{\sum_{y=0}^{\infty}s\left(\mu_0,\frac{\mu_0}{\sigma_0^2},y\right)}\right|$$

$$= \varepsilon_1(\mu_0,\sigma_0^2)$$

which trivially yields $|\mathbb{E}[Z] - \mu_0| \le \varepsilon_1(\mu_0,\sigma_0^2)$, as desired.

To evaluate $\text{Var}[Z]$, we must first compute the second derivative of $\mathscr{C}$:

$$\frac{\partial^2}{\partial \eta_1^2}\mathscr{C} = \frac{\partial^2 c}{\partial \mu^2}\left(\frac{\partial \mu}{\partial \eta_1}\right)^2 + \frac{\partial^2 \mu}{\partial \eta_1^2}\frac{\partial c}{\partial \mu}$$

$$= \frac{\mu^2}{\gamma^2}\frac{\partial}{\partial \mu}\left(\gamma^{\frac{3}{2}}\sum_{y=0}^{\infty}s(\mu,\gamma,y)\left(\frac{y-\mu}{\mu}\right)\right) + \frac{\mu}{\gamma^2}\left(\gamma^{\frac{3}{2}}\sum_{y=0}^{\infty}s(\mu,\gamma,y)\left(\frac{y-\mu}{\mu}\right)\right)$$

$$= \left(\frac{\mu^2}{\gamma^2}\right)\left(\gamma^{\frac{3}{2}}\right)\sum_{y=0}^{\infty}\left[s(\mu,\gamma,y)\gamma\left(\frac{y-\mu}{\mu}\right)^2 + s(\mu,\gamma,y)\left(\frac{-y}{\mu^2}\right)\right] + \left(\frac{\mu}{\gamma^2}\right)\left(\gamma^{\frac{3}{2}}\right)\sum_{y=0}^{\infty}s(\mu,\gamma,y)\left(\frac{y-\mu}{\mu}\right)$$

$$= \gamma^{-\frac{1}{2}}\left[\sum_{y=0}^{\infty}s(\mu,\gamma,y)\left(\gamma(y-\mu)^2 - y\right) + \sum_{y=0}^{\infty}s(\mu,\gamma,y)(y-\mu)\right]$$

$$= d(\mu,\gamma)$$

We now have an expression for the variance:

$$\mathrm{Var}[Z] = \frac{\exp\{\frac{\eta_1}{\eta_2}\}}{\eta_2} + \frac{\mathscr{C}(\eta_1,\eta_2)\frac{\partial^2}{\partial \eta_1^2}\mathscr{C}(\eta_1,\eta_2) - \left(\frac{\partial}{\partial \eta_1}\mathscr{C}(\eta_1,\eta_2)\right)^2}{\mathscr{C}(\eta_1,\eta_2)^2}$$

$$= \frac{\mu}{\gamma} + \frac{d(\mu,\gamma)\gamma^{\frac{1}{2}}\sum_{y=0}^{\infty}s(\mu,\gamma,y) - \gamma\left(\sum_{y=0}^{\infty}s(\mu,\gamma,y)(y-\mu)\right)^2}{\gamma\left(\sum_{y=0}^{\infty}s(\mu,\gamma,y)\right)^2}$$

So

$$\mathrm{Var}[Z] - \sigma_0^2 = \left(\frac{d\left(\mu_0,\frac{\mu_0}{\sigma_0^2}\right)\left(\frac{\mu_0}{\sigma_0^2}\right)^{\frac{1}{2}}\sum_{y=0}^{\infty}s\left(\mu_0,\frac{\mu_0}{\sigma_0^2},y\right) - \left(\frac{\mu_0}{\sigma_0^2}\right)\left(\sum_{y=0}^{\infty}s\left(\mu_0,\frac{\mu_0}{\sigma_0^2},y\right)(y-\mu_0)\right)^2}{\left(\frac{\mu_0}{\sigma_0^2}\right)\left(\sum_{y=0}^{\infty}s\left(\mu_0,\frac{\mu_0}{\sigma_0^2},y\right)\right)^2}\right)$$

which implies

$$\left|\mathrm{Var}[Z] - \sigma_0^2\right| = \left|\frac{d\left(\mu_0,\frac{\mu_0}{\sigma_0^2}\right)\left(\frac{\mu_0}{\sigma_0^2}\right)^{\frac{1}{2}}\sum_{y=0}^{\infty}s\left(\mu_0,\frac{\mu_0}{\sigma_0^2},y\right) - \left(\frac{\mu_0}{\sigma_0^2}\right)\left(\sum_{y=0}^{\infty}s\left(\mu_0,\frac{\mu_0}{\sigma_0^2},y\right)(y-\mu_0)\right)^2}{\left(\frac{\mu_0}{\sigma_0^2}\right)\left(\sum_{y=0}^{\infty}s\left(\mu_0,\frac{\mu_0}{\sigma_0^2},y\right)\right)^2}\right|$$

$$= \varepsilon_2(\mu_0,\sigma_0^2)$$

yielding $\left|\mathrm{Var}[Z] - \sigma_0^2\right| \le \varepsilon_2(\mu_0,\sigma_0^2)$, as desired.

$\square$

**Remark 1.** *The MDFs in the proposition yield near-zero values for most target means and variances, implying that in most settings, the Double Poisson can be considered a distribution with unrestricted variance.*

We now study the bounds obtained in Proposition A.2 empirically. In Figure 7 we plot the error, $\epsilon$, incurred via Efron's estimates on a grid of target means and variances, using 100th partial sums. Both $\epsilon_1$ and $\epsilon_2$ are effectively 0 for nearly all values of $\mu_0$ and $\sigma_0^2$. The only exception is for very small values of the target mean, $\mu_0 \to 0$; when the target variance is subsequently large, we see an increase in both $\epsilon_1$ and $\epsilon_2$. Under these rare conditions, the approximations begin to deteriorate.

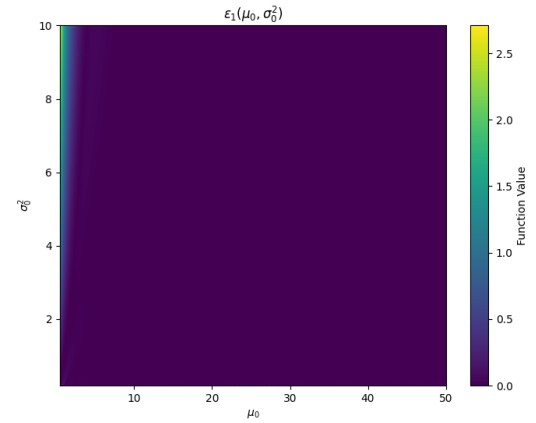

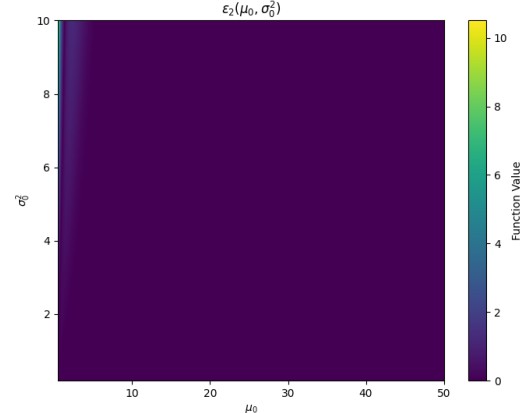

(a) Grid of values for $\epsilon_1(\mu_0, \sigma_0^2)$, the moment-deviation function for the Double Poisson mean.

(b) Grid of values for $\epsilon_2(\mu_0, \sigma_0^2)$, the moment-deviation function for the Double Poisson variance.

Figure 7: We plot the error, $\epsilon$, incurred via Efron's approximations of the Double Poisson mean and variance, on a grid of target values for these moments. Values were obtained using 100th partial sums.

### A.4. The effect of $\gamma$ initialization

Another potential method for avoiding the loss attenuation trap (see Section 3.3) is to initialize $\gamma$ to some high value (via the initial bias in the output head). This essentially forces the network to begin training with low uncertainty values. We investigate the effectiveness of this method by initializing $\gamma$ to various high values at the start of training. We train on the same dataset as in Figure 6. Results with differing $\gamma$ initializations are presented in Figure 8. Rather than improve mean fit, we observe that this strategy actually hurts overall convergence to the true function. The best performance comes, in fact, when $\log(\gamma)$ is initialized close to zero. Note that despite higher initialization of $\gamma$, the point to the far right of the data is still "explained" via low $\gamma$ (high uncertainty). Interestingly, the $\beta$ modification can help us avoid the impact of such poor initialization. In Figure 9, we mirror the experimental setup of Figure 8 but set $\beta = 1$ and train with $\beta$-NLL. In all but the most extreme case ($\log(\gamma_0) = 5$), $\beta$ allows us to recover the true function.

## B. Additional Experimental Details

### B.1. Description of Datasets

In this section, we provide a brief description of each dataset used in our experiments and define the associated network architectures. For further details, we refer the reader to Appendix B.3 and our source code.

**Length of Stay** Regression models on `Length of Stay` (Microsoft, 2016) are tasked with forecasting the number of days between initial admission and discharge for a given hospital patient, conditioned on a collection of health metrics and indicators about the specific facility providing the treatment. This is inherently a count prediction problem, since resource allocation and billing typically discretize time spent in a hospital. We train small MLPs to output the parameters of a predictive distribution over the number of days a patient is expected to stay.

**COCO-People** We introduce an image regression task on the `person` class of MS-COCO (Lin et al., 2014), which we call `COCO-People`. In this dataset, the task is to predict the number of people in each image. Each model we evaluate on `COCO-People` is trained with a small MLP on top of the pooled output from a ViT-B-16 backbone (Wu et al., 2020)).

**Inventory** This dataset comprises an inventory counting task (Jenkins et al., 2023), where the goal is to predict the number of objects on a retail shelf from an input point cloud (see Figure 32 in Appendix D.3 for an example). For this task, we adapt CountNet3D (Jenkins et al., 2023) for probabilistic regression.

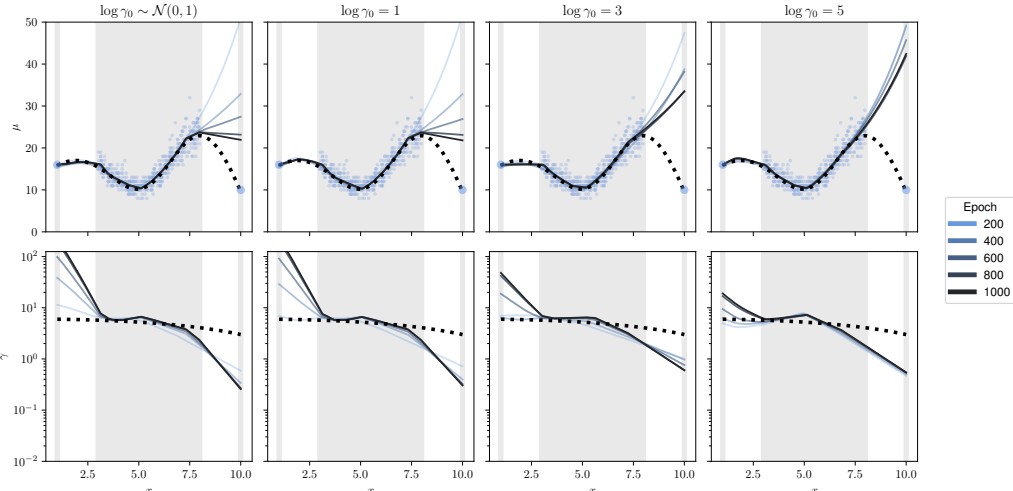

Figure 8: Results of training on a dataset with isolated points when we initialize $\gamma$ for a standard DDPN model to a high value (via the initial bias in the output head). Despite forcing low uncertainty at the beginning of training, the model still ignores difficult points and converges to a suboptimal solution. In fact, the smaller we force the uncertainty to be initially (through high $\gamma$ values), the worse our eventual fit is.

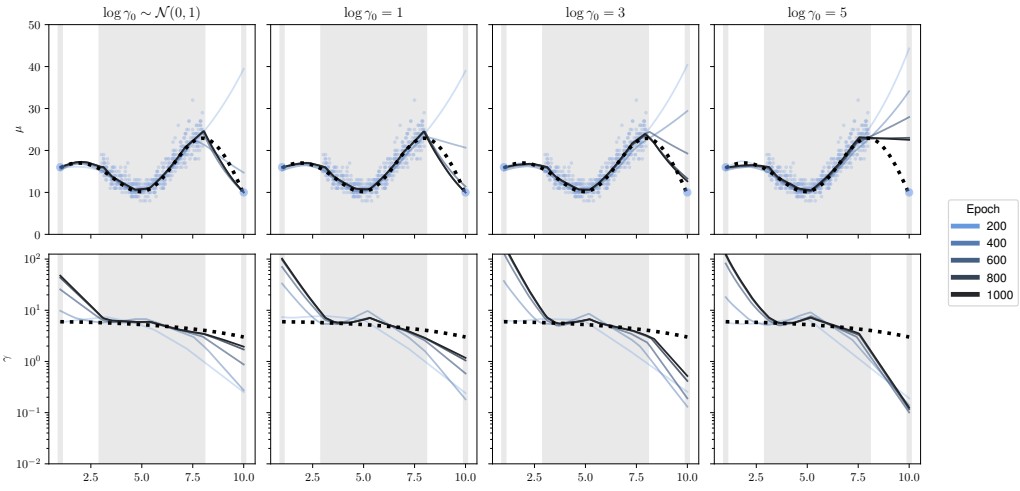

Figure 9: Rerun of the experiment depicted in Figure 8 where we train with $\beta_{1.0}$-NLL instead of vanilla NLL. Interestingly, we find that $\beta$ allows the model to recover from poor weight initializations.

**Amazon Reviews** We model user ratings from the "Patio, Lawn, and Garden" split of a collection of Amazon reviews (Ni et al., 2019). The objective in this task is to predict the discrete review value (1-5 stars) from an input text sequence, which historically has been addressed with Gaussian NLL (Mnih & Salakhutdinov, 2007; Koren et al., 2009). All text regressors consist of a small MLP on top of a DistilBert backbone (Sanh et al., 2019).

## B.2. Evaluation Metrics

In this section, we provide a detailed definition of each metric we employ for evaluation in our experiments. We also provide context for our choices around which metrics to use (when relevant).

### B.2.1. MEAN ABSOLUTE ERROR

The Mean Absolute Error (MAE) quantifies how closely a model's point predictions align with the ground truth across a dataset. In the probabilistic setting, we use the mode of the predictive distribution as our point prediction. Given a dataset $\{\mathbf{x}_i, y_i\}_{i=1}^n$ and model $\mathbf{f}_{\boldsymbol{\Theta}}$, the MAE is then computed as:

$$\text{MAE}(\mathbf{f}_{\boldsymbol{\Theta}}, \{\mathbf{x}_i, y_i\}_{i=1}^n) = \frac{1}{n}\sum_{i=1}^n \left| y_i - \underset{z \in \text{support}\{y_i\}}{\text{argmax}} \, p(y_i = z \mid \mathbf{f}_{\boldsymbol{\Theta}}(\mathbf{x}_i)) \right|$$

### B.2.2. CONTINUOUS RANKED PROBABILITY SCORE

The Continuous Ranked Probability Score (CRPS) is a strictly proper scoring rule that quantifies how aligned predictive distributions (sometimes referred to as "forecasts") are with ground-truth observations (Matheson & Winkler, 1976; Gneiting & Raftery, 2007). For a specific predictive cumulative density function (CDF), $\hat{F}_i$, and a ground-truth regression target $y_i$ with CDF $F_i$, it is defined as follows:

$$\begin{aligned}\text{CRPS}(\hat{F}_i, y_i) &= \int_{-\infty}^{\infty} [\hat{F}_i(z) - F_i(z)]^2 \, dz \\ &= \int_{-\infty}^{\infty} [\hat{F}_i(z) - \mathbb{1}_{\{z \geq y_i\}}]^2 \, dz \\ &= \int_{-\infty}^{y_i} \hat{F}_i(z)^2 \, dz + \int_{y_i}^{\infty} [\hat{F}_i(z) - 1]^2 \, dz \end{aligned}$$

Note that the second equality arises because $y_i$, once observed, is deterministic (its probability density is a point mass at the observed value). When the predictive distribution is discrete, this integral further reduces to the summation:

$$\sum_{z=0}^{y_i-1} \hat{F}_i(z)^2 dz + \sum_{z=y_i}^{\infty} [\hat{F}_i(z) - 1]^2$$

CRPS can also be computed from deterministic forecasts (i.e. those obtained from models that output only a point prediction). In this case, it is a known fact that CRPS reduces to the mean absolute error (Gneiting & Raftery, 2007), which aids interpretability. In our experiments, when we report CRPS, we take the average value across all test points (since CRPS is defined point-wise).

### B.2.3. MEDIAN PRECISION

Gneiting et al. (2007) observe that a good condition for a well-fit probabilistic model is "sharpness subject to calibration". In other words, our model must produce forecasts that exhibit statistical consistency with observations while also concentrating as much mass as possible around the true value of a regression target. Although sharpness alone is not a useful metric for quantifying probabilistic fit, measuring this value can provide helpful insights into the typical shape of a model's predictions. To quantify sharpness for a model/dataset pair, we use the Median Precision (MP), where precision is defined as $\lambda_i = \text{Var}[y_i | \mathbf{f}_{\boldsymbol{\Theta}}(\mathbf{x}_i)]^{-1}$. We employ the median instead of the mean as a summary statistic to increase robustness to outliers

(when the predictive variance is very small, the precision value blows up). In Appendix B.4.1, we report MP values for each model benchmarked in Section 4.3.

### B.2.4. On the Omission of Other Calibration Metrics

In this work, we choose not to quantify calibration via common metrics such as negative log likelihood (NLL) and expected calibration error (ECE). This is primarily due to the unique challenge of comparing probability forecasts across models with various distributional assumptions. NLLs obtained for continuous distributions are computed from probability *densities*, whereas discrete distributions output probability *masses*. These are fundamentally different quantities that cannot be directly compared. Meanwhile, the regression form of ECE (Kuleshov et al., 2018) has recently been shown to favor continuous models over discrete ones due to an implicit assumption about uniformity of the probability integral transform (Young & Jenkins, 2024). Other work suggests that ECE may not be a reliable indicator of true calibration in the first place, since it is a marginal measure (Song et al., 2019; Marx et al., 2024) and is sharply discontinuous as a function of the predictor — arbitrarily small perturbations to a model can cause large jumps in ECE (Błasiok et al., 2023).

We note that CRPS (see Appendix B.2.2), which is defined solely in terms of the distance between a predicted and ground-truth CDF and is a strictly proper scoring rule, overcomes the discrete vs. continuous comparison challenge and provides a reliable indicator of probabilistic fit. Thus, we include it as the main measure of calibration in our experiments (Section 4.3).

### B.2.5. Out-of-Distribution Classification Metrics

We use standard binary classification metrics to characterize each model's ability to identify out-of-distribution inputs, in line with Hendrycks & Gimpel (2016), Alemi et al. (2018), and Ren et al. (2019).

**AUROC**    The Area Under the Receiver Operating Characteristic Curve (AUROC) evaluates the performance of a binary classifier across a range of thresholds. In our out-of-distribution (OOD) experiment, we define these thresholds based on different expected false positive rates, i.e., $\alpha \in [0, 1]$. For each threshold, we plot the true positive rate (the percentage of true OOD samples correctly identified) against the false positive rate (the percentage of in-distribution samples mistakenly classified as OOD). This produces a curve whose integrated area provides a comprehensive measure of the classifier's performance. In this context, the AUROC can be interpreted as the probability that, given a random in-distribution (ID) and OOD sample, the model assigns a higher predictive variance to the OOD sample than the ID sample. Thus, higher values of AUROC indicate more useful classifiers. A score of 0.5 corresponds to random guessing. For further details, see Marzban (2004).

**AUPR**    The Area Under the Precision-Recall Curve (AUPR) is another metric for evaluating classifier performance, analogous to AUROC, but tailored for imbalanced datasets. Like AUROC, it involves computing the integral under a curve parameterized by $\alpha$. This curve plots precision (the proportion of OOD classifications that were correct) against recall (the proportion of all true OOD instances that were correctly classified), illustrating the trade-off between these two metrics. Higher values are better. A model relying solely on class proportions in the data would achieve an AUPR equal to $P(X \text{ is OOD})$, which in our experiment is 0.224.

**FPR80**    The False Positive Rate at 80% True Positive Rate (FPR80) quantifies the proportion of in-distribution (ID) samples misclassified as out-of-distribution (OOD) when the classification threshold $\alpha$ is set to achieve a True Positive Rate (TPR) of 80% (Ren et al., 2019). This metric highlights the classifier's ability to minimize false positives while maintaining high sensitivity to OOD samples. Lower values indicate better performance, with a perfect classifier achieving an FPR80 of zero.

### B.3. Training Details

In all experiments, instead of using the final set of weights achieved during training with a particular technique, we select the weights associated with the best average loss on a held-out validation set. This can be viewed as a form of early stopping.

The ReLU (Fukushima, 1969) activation is exclusively used for all MLPs. No dropout or batch normalization is applied. We employ the AdamW optimizer (Loshchilov & Hutter, 2017) for all training procedures, and anneal the initial learning rate to 0 following a cosine schedule (Loshchilov & Hutter, 2016).

The hyperparameter settings we report are chosen such that each model converges to a stable training loss basin with minimal

overfitting (as quantified by validation loss). To ensure fairness across all models, we use one set of training hyperparameters (including choice of underlying architecure) for each specific dataset.

In addition to the details we provide in this section, we refer the reader to our source code [3], where configuration files for each model benchmarked can be found. The corresponding author can be contacted for access to our pre-generated splits of each dataset.

### B.3.1. SIMULATION EXPERIMENT (INTRODUCTORY FIGURE)

To produce Figure 1, we generate a dataset with the following procedure: First, we sample $x$ from a uniform distribution, $x \sim \texttt{Uniform}(0, 2\pi)$. Next, we draw an initial proposal for $y$ from a conflation (Hill, 2011) of five identical Poissons, each with rate parameterized by $\lambda(x) = 10\sin(x) + 10$. We scale $y$ by $-1$ and shift it by $+30$ to achieve high dispersion at low counts and under-dispersion at high counts while maintaining nonnegativity. We generate 800 points from this process for the training split, as well as 100 for validation and 100 for evaluation.

The models that make up the ensembles we report in this figure are small MLPs (with layers of width `[128, 128, 128, 64]`). We train each network for 200 epochs on the CPU of a 2021 MacBook Pro with a batch size of 32 and an initial learning rate of $10^{-3}$. We set weight decay to $10^{-5}$.

### B.3.2. LENGTH OF STAY

Each model benchmarked for `Length of Stay` is a small MLP with layer widths `[128, 128, 128, 64]`. Models are trained for 15 epochs on a 2021 MacBook Pro CPU with a batch size of 128, an initial learning rate of $10^{-4}$, and a weight decay value of $10^{-4}$.

The `Length of Stay` dataset has been open-sourced by Microsoft under the MIT license. It has 100,000 rows, which we divide into a 80/10/10 train/val/test split using a fixed random seed.

Licensing information for `Length of Stay` can be found at `https://github.com/microsoft/ r-server-hospital-length-of-stay/blob/master/Website/package.json`.

The full dataset can be downloaded from `https://raw.githubusercontent.com/microsoft/ r-server-hospital-length-of-stay/refs/heads/master/Data/LengthOfStay.csv`.

### B.3.3. COCO-PEOPLE

We train Vision Transformers (Wu et al., 2020) on `COCO-People`, passing their pooled patch representations to a two-layer MLP regression head with layer widths of `[384, 256]`. All weights are fine-tuned from the `vit-base-patch16-224-in21k` checkpoint (Deng et al., 2009). We set the initial learning rate to $10^{-4}$ and use a weight decay of $10^{-3}$. We train with an effective batch size of 256 in a distributed fashion, using an on-prem machine with 2 Nvidia GeForce RTX 4090 GPUs. Images are normalized with the ImageNet (Deng et al., 2009) pixel means and standard deviations and augmented during training with the `AutoAugment` transformation (Cubuk et al., 2018). Training is done with BFloat 16 Mixed Precision.

The `COCO` dataset from which we form the `COCO-People` subset is distributed via the CCBY 4.0 license. It can be accessed at `https://cocodataset.org/#home`. Our subset has 64,115 images in total, which we divide into a 70/10/20 train/val/test split.

### B.3.4. INVENTORY

For the `Inventory` task, we use a modified version of CountNet3D (Jenkins et al., 2023) that ouputs the parameters of a probability distribution instead of regressing the mean directly. We train models for 50 epochs with an effective batch size of 16, an initial learning rate of $10^{-3}$, and weight decay of $10^{-5}$. Training is performed on an internal cluster of 4 NVIDIA GeForce RTX 2080 Ti GPUs.

The `Inventory` dataset is made available to us via an industry collaboration and is not publicly accessible.

---

[3]`https://github.com/delicious-ai/ddpn`

### B.3.5. AMAZON REVIEWS

We fine-tune a DistilBert Base Cased (Sanh et al., 2019) backbone with a `[384, 256]` MLP regression head for the `Amazon Reviews` task. To avoid overfitting, we freeze all but the final transformer block in the backbone. The regression head is not frozen. All networks are trained for 10 epochs across 2 on-prem Nvidia GeForce RTX 4090 GPUs with an effective batch size of 2048. We use an initial learning rate of $10^{-4}$ and a weight decay of $10^{-5}$, and run computations with BFloat 16 Mixed Precision.

`Amazon Reviews` is publicly available at `https://cseweb.ucsd.edu/~jmcauley/datasets/amazon_v2/`. The "Patio, Lawn, and Garden" subset we employ in this work is hosted at `https://datarepo.eng.ucsd.edu/mcauley_group/data/amazon_v2/categoryFilesSmall/Patio_Lawn_and_Garden.csv`. Our subset contains 793,966 total reviews, which we divide into a 70/10/20 train/val/test split.

## B.4. Additional Results

### B.4.1. SHARPNESS

To provide a notion of the sharpness of predictive distributions produced by each model we benchmark, we measure and report the median precision (MP) across all datasets in Table 4. See Appendix B.2.3 for more details about how we compute this metric. Higher values correspond to sharper distributions (those with lower predictive variance). We note that sharpness alone is not an indication of good probabilistic fit; thus we do not mark "winners" in this metric. We simply provide sharpness values to facilitate improved understanding of each technique's behavior on our datasets.

Table 4: Median Precision (MP) values across main experiments.

|  |  | Length of Stay | COCO-People | Inventory | Reviews |
|---|---|---|---|---|---|
| Aleatoric Only | Poisson DNN | 0.262 (0.00) | 0.437 (0.01) | 0.263 (0.01) | 0.211 (0.00) |
|  | NB DNN | 0.265 (0.00) | 0.392 (0.04) | 0.261 (0.01) | 0.211 (0.00) |
|  | Gaussian DNN | 2.031 (0.09) | 0.913 (0.43) | 0.689 (0.02) | 3.833 (0.28) |
|  | Faithful Gaussian | 1.258 (0.09) | 0.647 (0.05) | 1.072 (0.13) | 3.907 (0.04) |
|  | Natural Gaussian | 1.456 (0.07) | 0.745 (0.20) | 0.713 (0.02) | 4.545 (0.29) |
|  | $\beta_{0.5}$-Gaussian | 2.125 (0.07) | 2.174 (0.18) | 1.011 (0.07) | 4.604 (0.45) |
|  | $\beta_{1.0}$-Gaussian | 1.646 (0.21) | 1.558 (0.09) | 0.814 (0.06) | 3.935 (0.04) |
|  | DDPN (ours) | 2.663 (0.14) | 0.915 (0.30) | 0.680 (0.04) | 3.960 (0.13) |
|  | $\beta_{0.5}$-DDPN (ours) | 2.184 (0.14) | 1.006 (0.13) | 0.818 (0.04) | 3.642 (0.18) |
|  | $\beta_{1.0}$-DDPN (ours) | 1.565 (0.11) | 1.986 (0.04) | 0.690 (0.03) | 3.806 (0.04) |
| Aleatoric + Epistemic | Poisson DNN | 0.261 | 0.413 | 0.253 | 0.210 |
|  | NB DNN | 0.263 | 0.354 | 0.252 | 0.210 |
|  | Gaussian DNN | 1.933 | 0.548 | 0.631 | 3.643 |
|  | Faithful Gaussian | 1.211 | 0.592 | 0.907 | 3.927 |
|  | Natural Gaussian | 1.386 | 0.571 | 0.639 | 4.217 |
|  | $\beta_{0.5}$-Gaussian | 2.003 | 1.785 | 0.828 | 4.362 |
|  | $\beta_{1.0}$-Gaussian | 1.531 | 1.360 | 0.734 | 3.819 |
|  | DDPN (ours) | 2.441 | 0.626 | 0.623 | 3.694 |
|  | $\beta_{0.5}$-DDPN (ours) | 2.046 | 0.821 | 0.709 | 3.645 |
|  | $\beta_{1.0}$-DDPN (ours) | 1.463 | 1.680 | 0.620 | 3.569 |

### B.4.2. GLM RESULTS ON LENGTH OF STAY

Since `Length of Stay` is tabular, we may compare GLM baselines in addition to the neural networks benchmarked in Section 4.3. We train and evaluate a Poisson Generalized Linear Model (GLM), a Negative Binomial GLM, and a Double Poisson GLM (Efron, 1986; Zhu, 2012; Zou et al., 2013; Aragon et al., 2018; Toledo et al., 2022). As in Table 2, we train 5 independent models with each technique and report the mean and standard error of both MAE and CRPS across all trials under the "Aleatoric Only" header. We also provide the median precision values as in Table 4. Although GLMs are not deep networks, we may ensemble their predictions in a similar fashion as deep ensembles (DEs) to quantify epistemic uncertainty.

Table 5: GLM results for the Length of Stay dataset.

| | Aleatoric Only | | | Aleatoric + Epistemic | | |
|---|---|---|---|---|---|---|
| | MAE ($\downarrow$) | CRPS ($\downarrow$) | MP | MAE ($\downarrow$) | CRPS ($\downarrow$) | MP |
| Poisson GLM | 1.284 (0.08) | 0.882 (0.06) | 0.319 (0.01) | 1.256 | 0.850 | 0.309 |
| NB GLM | 3.783 (0.27) | 1.692 (0.14) | 0.241 (0.00) | 3.906 | 1.685 | 0.239 |
| DP GLM | 1.320 (0.15) | 0.894 (0.08) | 0.543 (0.44) | 1.185 | 0.829 | 0.340 |

We do so, and report these results under the "Aleatoric + Epistemic" header.

The limited flexibility of linear models appears to hinder the GLMs we evaluate, as they struggle with both mean fit and calibration when compared to their respective neural network counterparts.

## C. Proofs and Derivations

In this section we provide full proofs of the propositions presented in the main body of the paper, along with derivations of key equations (excepting the Double Poisson NLL derivation, which is located in Appendix A.1)

### C.1. Proof of Proposition 2.3

*Proof.* We will first demonstrate that Gaussian regressors are fully heteroscedastic. Since they model both the predicted mean $\mu_i$ and variance $\sigma_i^2$ conditionally, all we have to show is that the Normal distribution exhibits unrestricted variance. To see this, note that if $Z$ is distributed normally, if we fix $\mu$ for any $\mu \in \mathbb{R}$, then for an arbitrary $\sigma^2 \in (0, \infty)$, the parameter setting $\boldsymbol{\psi} = [\mu, \sigma^2]^T$ achieves $\text{Var}[Z] = \sigma^2$. Thus the Normal distribution has unrestricted variance, and Gaussian regressors are indeed fully heteroscedastic.

To show that Poisson and Negative Binomial regressors are not fully heteroscedastic, we will prove that their respective output distributions do not have unrestricted variance, i.e. there exists at least one $\sigma^2 \in (0, \infty)$ such that no setting of parameters $\boldsymbol{\psi}$ can achieve this variance when the expected value is fixed.

If we say $Z$ is Poisson, then for any $\lambda > 0$, fixing $\mathbb{E}[Z] = \lambda$ requires $Z \sim \text{Poisson}(\lambda)$, i.e. $\boldsymbol{\psi} = [\lambda]$. For any $\sigma^2 \in (0, \infty) \setminus \{\lambda\}$, we see that there is no setting of $\boldsymbol{\psi}$ such that $\text{Var}[Z] = \sigma^2$, since $\text{Var}[Z] = \mathbb{E}[Z] = \lambda$. So the Poisson distribution does not have unrestricted variance.

The Negative Binomial case is similar. We employ the common $(r, p)$ parametrization, though other alternatives are possible. Recall that in this parametrization, $r > 0$ is the number of successful trials until an experiment is stopped, and $p \in (0, 1)$ is the probability of success in each trial of the experiment. Pick any $\mu > 0$. If $Z$ is Negative Binomial, then fixing $\mathbb{E}[Z] = \mu$ means that $\boldsymbol{\psi} = [r, p]^T$ must satisfy $\mu = \frac{r(1-p)}{p}$. Now choose any $\sigma^2 \in (0, \mu)$. For $\text{Var}[Z] = \sigma^2$ to be true, $\boldsymbol{\psi}$ must satisfy $\sigma^2 = \frac{r(1-p)}{p^2}$. Noting that we chose $\sigma^2$ such that $\sigma^2 < \mu$, this implies $\frac{r(1-p)}{p^2} < \frac{r(1-p)}{p}$. Multiplying both sides by the positive quantity $\frac{p^2}{r(1-p)}$ yields $1 < p$. But $p < 1$ by definition. Thus we have identified a range of $\sigma^2$ values such that there exists no $\boldsymbol{\psi}$ that can satisfy $\text{Var}[Z] = \sigma^2$ if $\mathbb{E}[Z] = \mu$. So the Negative Binomial distribution does not have unrestricted variance.

We conclude that any regression model outputting either the Poisson or Negative Binomial distribution is not fully heteroscedastic.

$\square$

### C.2. Proof of Proposition 3.1

*Proof.* We assume the first and second moment approximations proposed by Efron (1986): $\mathbb{E}[Z] = \mu$ and $\text{Var}[Z] = \frac{\mu}{\gamma}$. To see that DDPN is fully heteroscedastic, note that for any input $\mathbf{x_i}$, DDPN predicts $\log(\hat{\gamma}_i) = \mathbf{w}_\gamma^T \mathbf{z_i} + b_\gamma$, where $\mathbf{z_i}$ is a neural network's input-conditional hidden representation of $\mathbf{x}_i$. Since DDPN's predictive variance directly depends on $\hat{\gamma}_i$ (which we just showed depends on $\mathbf{x}_i$), all we have left to demonstrate is that the Double Poisson distribution has unrestricted variance. To see this, note that if $Z$ is Double Poisson with expectation $\mu$ (where $\mu > 0$), then for any $\sigma^2 \in (0, \infty)$, choosing

$\psi = [\mu, \frac{\mu}{\sigma^2}]^T$ implies $\text{Var}[Z] = \frac{\mu}{\frac{\mu}{\sigma^2}} = \sigma^2$. Thus, DDPN is fully heteroscedastic. $\qquad \square$

### C.3. Proof of Proposition 3.3

*Proof.* Suppose that we have a loss function $\mathscr{L}(\hat{\mu}_i, \hat{\phi}_i)$ that can be written in the form $d(\hat{\phi}_i) + a(\hat{\phi}_i)r(\hat{\mu}_i, y_i)$ as stated in Definition 3.2. We have

$$\lim_{\hat{\phi}_i \to \infty} \frac{a(\hat{\phi}_i)r(\hat{\mu}_i, y_i)}{d(\hat{\phi}_i)} = r(\hat{\mu}_i, y_i) \lim_{\hat{\phi}_i \to \infty} \frac{a(\hat{\phi}_i)}{d(\hat{\phi}_i)}$$
$$= r(\hat{\mu}_i, y_i)(0) \qquad\qquad \text{(since } a \to 0 \text{ and } d \to \infty)$$
$$= 0$$

where for the second equality, we rely on the conditions imposed for $a$ and $d$ in Definition 3.2. $\qquad \square$

### C.4. Proof of Proposition 3.4

*Proof.* To see that DDPN exhibits learnable loss attenuation and conforms with Definition 3.2, recall the loss function from Equation 2:

$$\mathscr{L} = -\frac{\log \hat{\gamma}_i}{2} + \hat{\gamma}_i \hat{\mu}_i - \hat{\gamma}_i y_i (1 + \log \hat{\mu}_i - \log y_i) \tag{5}$$

Because DDPN is expressed in terms of inverse dispersion, $\gamma_i = \frac{1}{\phi_i}$, we can substitute $\frac{1}{\phi_i}$ in for $\gamma_i$ to re-parameterize the loss function in terms of dispersion:

$$\mathscr{L} = -\frac{1}{2}\log\frac{1}{\hat{\phi}_i} + \frac{\hat{\mu}_i}{\hat{\phi}_i} - \frac{y_i}{\hat{\phi}_i}(1 + \log \hat{\mu}_i - \log y_i) \tag{6}$$

We can rearrange terms to put $\mathscr{L}$ in the standard form with dispersion penalty $d(\phi_i)$, attenuation factor $a(\phi_i)$, and residual penalty $r(\mu_i, y_i)$:

$$\mathscr{L} = -\frac{1}{2}\log\frac{1}{\hat{\phi}_i} + \frac{1}{\hat{\phi}_i}\left[\hat{\mu}_i - y_i(1 + \log\hat{\mu}_i - \log y_i)\right]$$
$$= -\frac{1}{2}\log\frac{1}{\hat{\phi}_i} + \frac{1}{\hat{\phi}_i}\left[(\hat{\mu}_i - y_i) - y_i(\log\hat{\mu}_i - \log y_i)\right]$$
$$= -\frac{1}{2}\log 1 + \frac{1}{2}\log\hat{\phi}_i + \frac{1}{\hat{\phi}_i}\left[(\hat{\mu}_i - y_i) - y_i(\log\hat{\mu}_i - \log y_i)\right]$$
$$= \frac{1}{2}\log\hat{\phi}_i + \frac{1}{\hat{\phi}_i}\left[(\hat{\mu}_i - y_i) - y_i(\log\hat{\mu}_i - \log y_i)\right]$$

Let $d(\hat{\phi}_i) = \frac{1}{2}\log\hat{\phi}_i$, $a(\hat{\phi}_i) = \frac{1}{\hat{\phi}_i}$, and $r(\hat{\mu}_i, y_i) = (\hat{\mu}_i - y_i) - y_i(\log\hat{\mu}_i - \log y_i)$. We have $d(\hat{\phi}_i) + a(\hat{\phi}_i)r(\hat{\mu}_i, y_i) = \log\hat{\phi}_i + \frac{1}{\hat{\phi}_i}\left[(\hat{\mu}_i - y_i) - y_i(\log\hat{\mu}_i - \log y_i)\right]$. Clearly, $d(\phi) = \log\hat{\phi}_i$ is monotonically *increasing* w.r.t $\hat{\phi}_i$ and $a(\phi_i) = \frac{1}{\hat{\phi}_i}$ is both monotonically *decreasing* w.r.t $\hat{\phi}_i$ and bounded from below by 0 (since $\hat{\phi}_i = \frac{1}{\hat{\gamma}_i} > 0$).

All that is left to show is that (i) $r(\hat{\mu}_i, y_i) = 0 \iff \hat{\mu}_i = y_i$ and (ii) $r$ is nonnegative:

(i) We first show that $\hat{\mu}_i = y_i \implies r(\hat{\mu}_i, y_i) = 0$. To see this, note that when $\hat{\mu}_i = y_i$, both the terms in our residual penalty, $(\hat{\mu}_i - y_i)$ and $y_i(\log\hat{\mu}_i - \log y_i)$, are 0. So their difference, $r$, must be 0.

To see that $r(\hat{\mu}_i, y_i) = 0 \implies \hat{\mu}_i = y_i$, we handle the case of $y_i = 0$ and $y_i > 0$ separately:

**Case 1** If $y_i = 0$, then $r = 0 \implies (\hat{\mu}_i - y_i) - y_i(\log \hat{\mu}_i - \log y_i) = 0$. But $y_i = 0$ cancels the second term in this difference, leaving us with $0 = (\hat{\mu}_i - y_i) \implies \hat{\mu}_i = y_i$.

**Case 2** If $y_i > 0$, we can make the substitution $u = \frac{\hat{\mu}_i}{y_i}$ and rewrite the residual penalty: $r = uy_i - y_i - y_i \log u = y_i(u - 1 - \log u)$. Since $y_i > 0$, this yields $r = 0 \implies (u - 1 - \log u) = 0$. We now claim that $h(u) = (u - 1 - \log u) = 0 \implies u = 1$. To see this, note that $h'(u) = 1 - \frac{1}{u}$ is negative when $u < 1$ and positive when $u > 1$. Since $h(u) = 0$ when $u = 1$, strictly increasing to the right of $u$, and strictly decreasing to the left of $u$, 0 is uniquely achieved by $h$ at $u = 1$. So $h = 0 \implies u = 1$. Thus, $u - 1 - \log u = 0 \implies u = 1 \implies \frac{\hat{\mu}_i}{y_i} = 1 \implies \hat{\mu}_i = y_i$.

In either case, we have $\hat{\mu}_i = y_i$. Since we have proved both directions, we conclude that $r(\hat{\mu}_i, y_i) = 0 \iff \hat{\mu}_i = y_i$.

(ii) We separately handle the cases of $y_i = 0$ and $y_i > 0$, and may now assume (by virtue of the iff) that $\hat{\mu}_i \neq y_i$:

**Case 1** If $y_i = 0$, then $r = \hat{\mu}_i > 0$, where the last inequality comes from the restrictions on the parameters of a Double Poisson distribution.

**Case 2** If $y_i > 0$, then substituting $u = \frac{\hat{\mu}_i}{y_i}$ as before yields $r = y_i(u - 1 - \log u)$, which is positive if $h(u) = (u - 1 - \log u)$ is positive. Recall that $h(u) = 0$ iff $u = 1$, and that $h$ is strictly decreasing to the left of $u$ and strictly increasing to the right of $u$. Since $h'(u) = 0$, this implies (by the first derivative test) that $u = 1$ is the unique minimizer of $h$, with $h(u) = 0$. But we only have $u = 1$ if $\hat{\mu}_i = y_i$, which we can assume is not the case because of (i). So $h > 0 \implies (u - 1 - \log u) > 0 \implies r > 0$.

In both cases, we are guaranteed that $r > 0$.

Thus Equation 2 conforms with Definition 3.2.

$\square$

### C.5. Computing the Mean and Variance of a Mixture Model

In Section 3.4 we describe how the ensembled predictive distribution is a uniform mixture of the $M$ members of the ensemble: $p(y_i|\mathbf{x}_i) = \frac{1}{M} \sum_{m=1}^{M} p(y_i|\mathbf{f}_{\Theta_m}(\mathbf{x}_i))$.

Letting $\mu_m = \mathbb{E}[y_i|\mathbf{f}_{\Theta_m}(\mathbf{x}_i)]$ and $\sigma_m^2 = \text{Var}[y_i|\mathbf{f}_{\Theta_m}(\mathbf{x}_i)]$, we can get the mean and variance of the predictive distribution as $\mathbb{E}[y_i|\mathbf{x}_i] = \frac{1}{M} \sum_{m=1}^{M} \mu_m$ and $\text{Var}[y_i|\mathbf{x}_i] = \sum_{m=1}^{M} \frac{\sigma_m^2 + \mu_m^2}{M} - \left(\sum_{m=1}^{M} \frac{\mu_m}{M}\right)^2$.

We note that the ensembling technique used throughout our work can be applied to form an ensemble from any collection of neural networks outputting a probability distribution, regardless of the specific parametric form (Marron & Wand, 1992).

### C.6. Decomposing the Predictive Variance of a Mixture Model

For the mixture model described above, we can decompose the predictive variance for a single regression target $y_i$ as follows:

$$\text{Var}[y_i|\mathbf{x}_i] = \sum_{m=1}^{M} \frac{\sigma_m^{2\,(i)} + \mu_m^{(i)2}}{M} - \left(\sum_{m=1}^{M} \frac{\mu_m^{(i)}}{M}\right)^2$$

$$= \left(\frac{1}{M} \sum_{m=1}^{M} \sigma_m^{2\,(i)}\right) + \left[\frac{1}{M} \sum_{m=1}^{M} \mu_m^{(i)2} - \left(\frac{1}{M} \mu_m^{(i)}\right)^2\right]$$

$$= \mathbb{E}_m[\sigma_m^{2\,(i)}] + \left[\mathbb{E}_m[\mu_m^{(i)2}] - (\mathbb{E}_m[\mu_m^{(i)}])^2\right]$$

$$= \mathbb{E}_m[\sigma_m^{2\,(i)}] + \text{Var}_m[\mu_m^{(i)}]$$

The first term in this sum, $\mathbb{E}_m[\sigma_m^{2\,(i)}]$, is the average predictive variance of each ensemble member and is often considered to be a proxy for aleatoric uncertainty, while $\text{Var}_m[\mu_m^{(i)}]$ represents the amount of disagreement between ensemble members regarding the predicted mean and is interpreted as epistemic uncertainty (Kendall & Gal, 2017).

# D. Additional Case Studies

## D.1. Example Predictive Distributions on `COCO-People`

For each baseline, Figures 10-19 depict the predictive distributions of one model trained under that framework on `COCO-People`, while 20-29 show the combined predictions of a 5-model ensemble. We select 3 instances from the test split and visualize outputs on each. A black star indicates the true count of people in the image.

### D.1.1. INDIVIDUAL MODELS

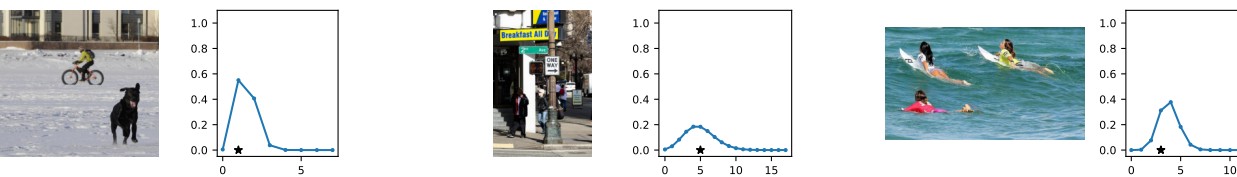

Figure 10: Predictive PMFs from a DDPN on samples from the test split of `COCO-People`.

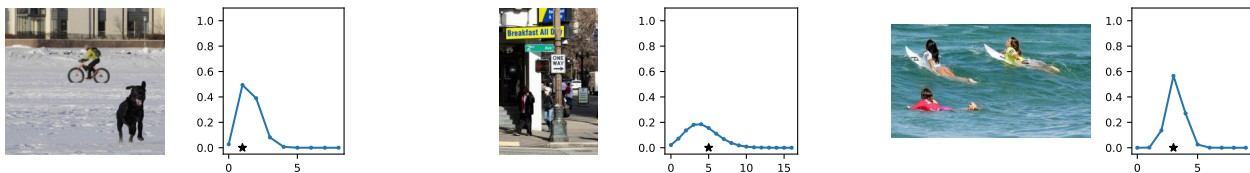

Figure 11: Predictive PMFs from a $\beta_{0.5}$-DDPN on samples from the test split of `COCO-People`.

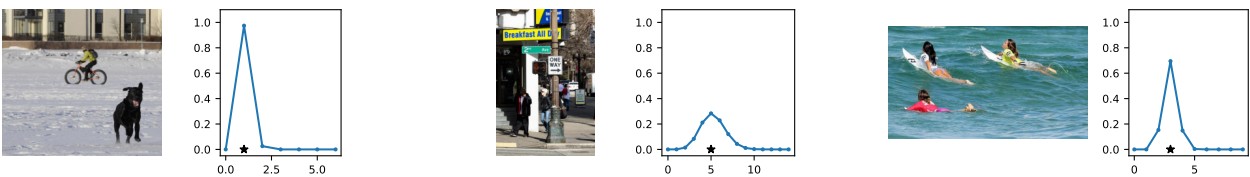

Figure 12: Predictive PMFs from a $\beta_{1.0}$-DDPN on samples from the test split of `COCO-People`.

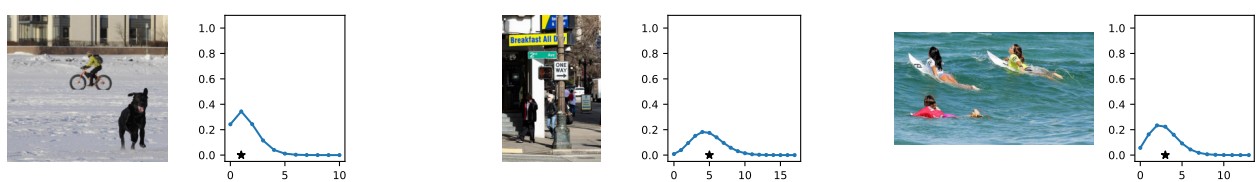

Figure 13: Predictive PMFs from a Poisson DNN on samples from the test split of `COCO-People`.

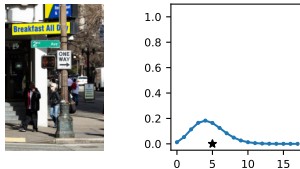 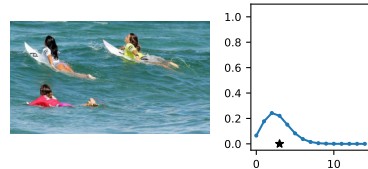

Figure 14: Predictive PMFs from a Negative Binomial DNN on samples from the test split of `COCO-People`.

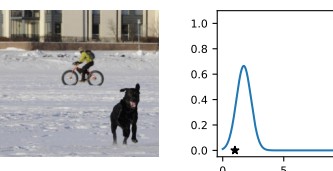 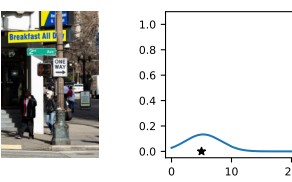 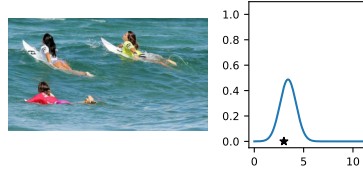

Figure 15: Predictive PMFs from a Gaussian DNN on samples from the test split of `COCO-People`.

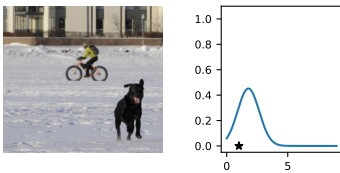 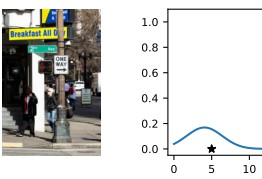 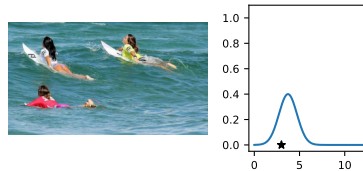

Figure 16: Predictive PMFs from a Natural Gaussian DNN on samples from the test split of `COCO-People`.

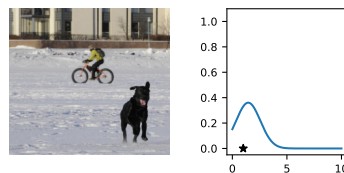 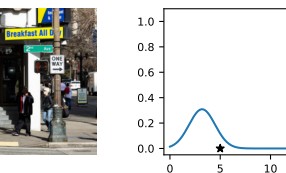 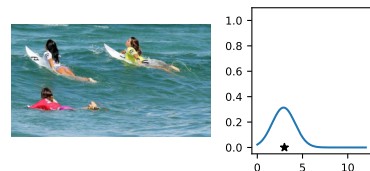

Figure 17: Predictive PMFs from a Faithful Gaussian DNN on samples from the test split of `COCO-People`.

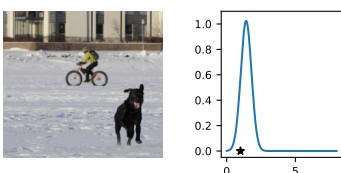 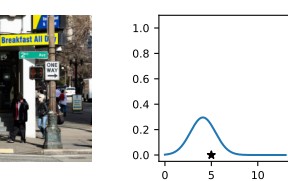 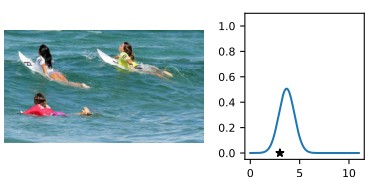

Figure 18: Predictive PMFs from a $\beta_{0.5}$-Gaussian DNN on samples from the test split of `COCO-People`.

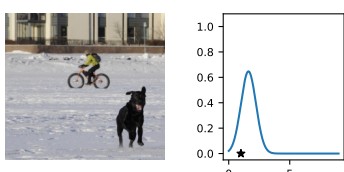 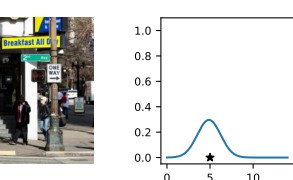 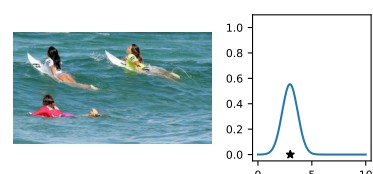

Figure 19: Predictive PMFs from a $\beta_{1.0}$-Gaussian DNN on samples from the test split of `COCO-People`.

D.1.2. ENSEMBLES

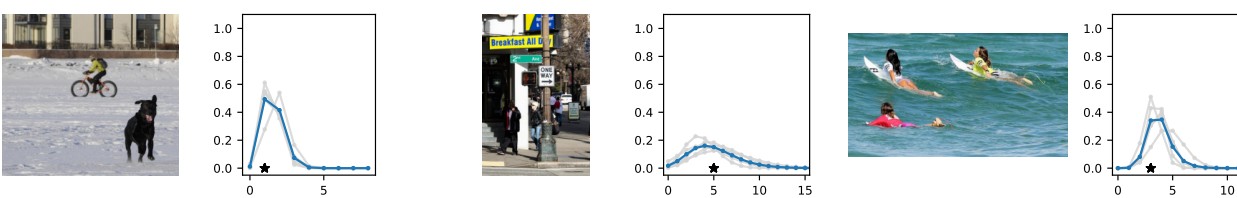

Figure 20: Predictive PMFs from a DDPN ensemble on samples from the test split of COCO-People.

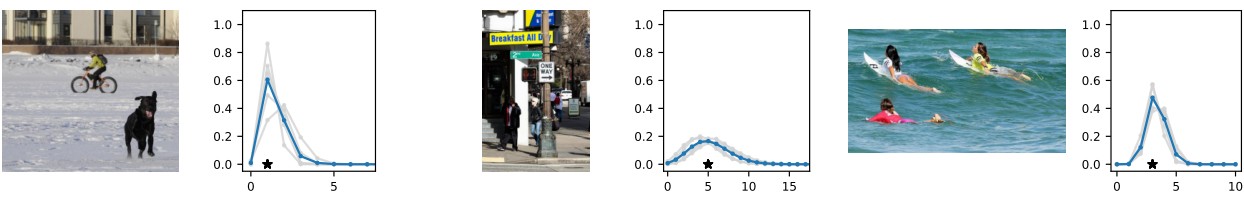

Figure 21: Predictive PMFs from a $\beta_{0.5}$-DDPN ensemble on samples from the test split of COCO-People.

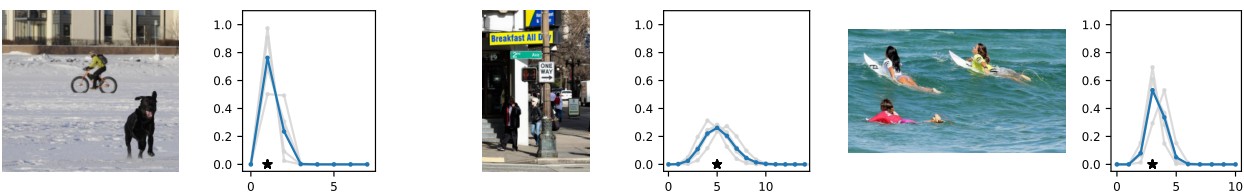

Figure 22: Predictive PMFs from a $\beta_{1.0}$-DDPN ensemble on samples from the test split of COCO-People.

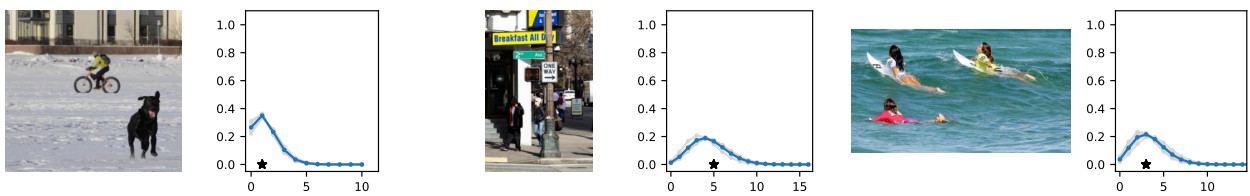

Figure 23: Predictive PMFs from a Poisson DNN ensemble on samples from the test split of COCO-People.

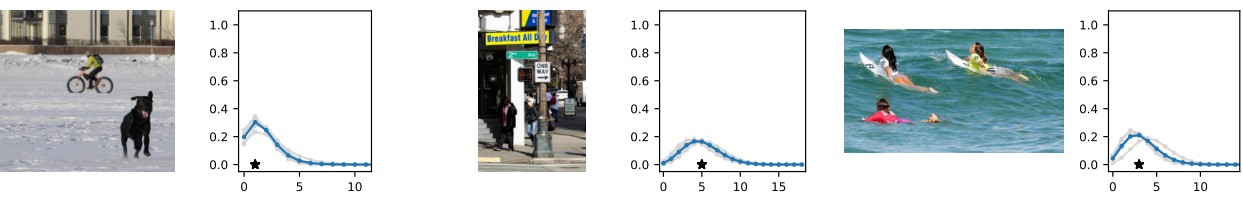

Figure 24: Predictive PMFs from a Negative Binomial DNN ensemble on samples from the test split of COCO-People.

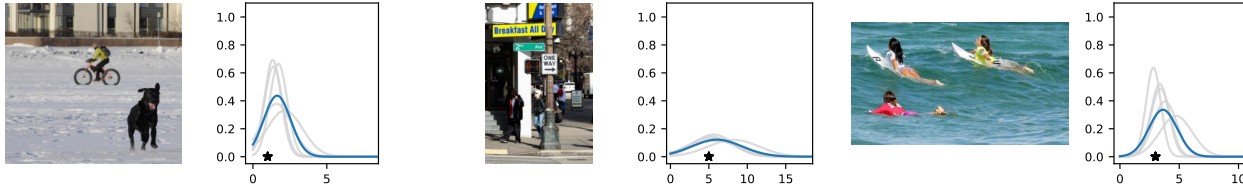

Figure 25: Predictive PMFs from a Gaussian DNN ensemble on samples from the test split of `COCO-People`.

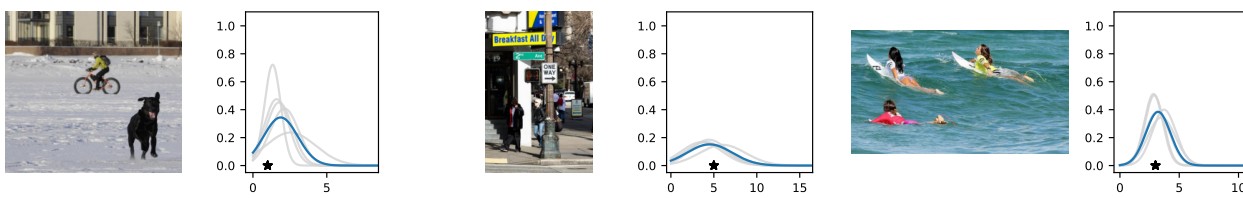

Figure 26: Predictive PMFs from a Natural Gaussian DNN ensemble on samples from the test split of `COCO-People`.

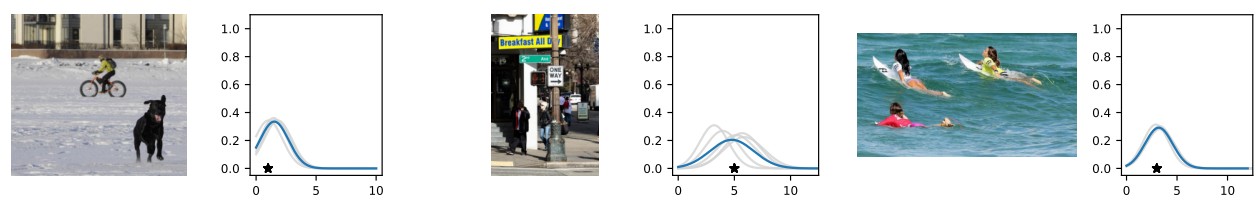

Figure 27: Predictive PMFs from a Faithful Gaussian DNN ensemble on samples from the test split of `COCO-People`.

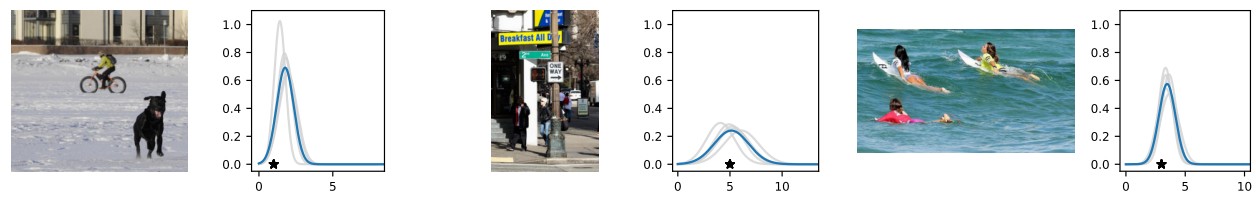

Figure 28: Predictive PMFs from a $\beta_{0.5}$-Gaussian DNN ensemble on samples from the test split of `COCO-People`.

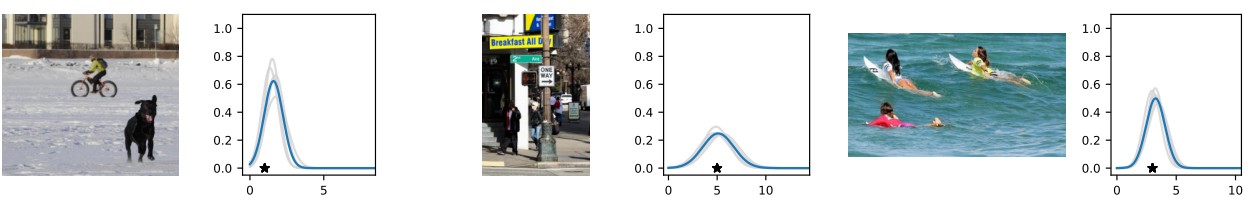

Figure 29: Predictive PMFs from a $\beta_{1.0}$-Gaussian DNN ensemble on samples from the test split of `COCO-People`.

## D.2. ID / OOD Predictive Distributions on `Amazon Reviews`

Figures 30 and 31 visualize the difference (or lack thereof) between ensemble predictions on in-distribution data (in this case, the test split of `Amazon Reviews`) and out-of-distribution data (verses from the King James Bible). We include predictive distributions for each ensemble baseline presented in Section 4.3.

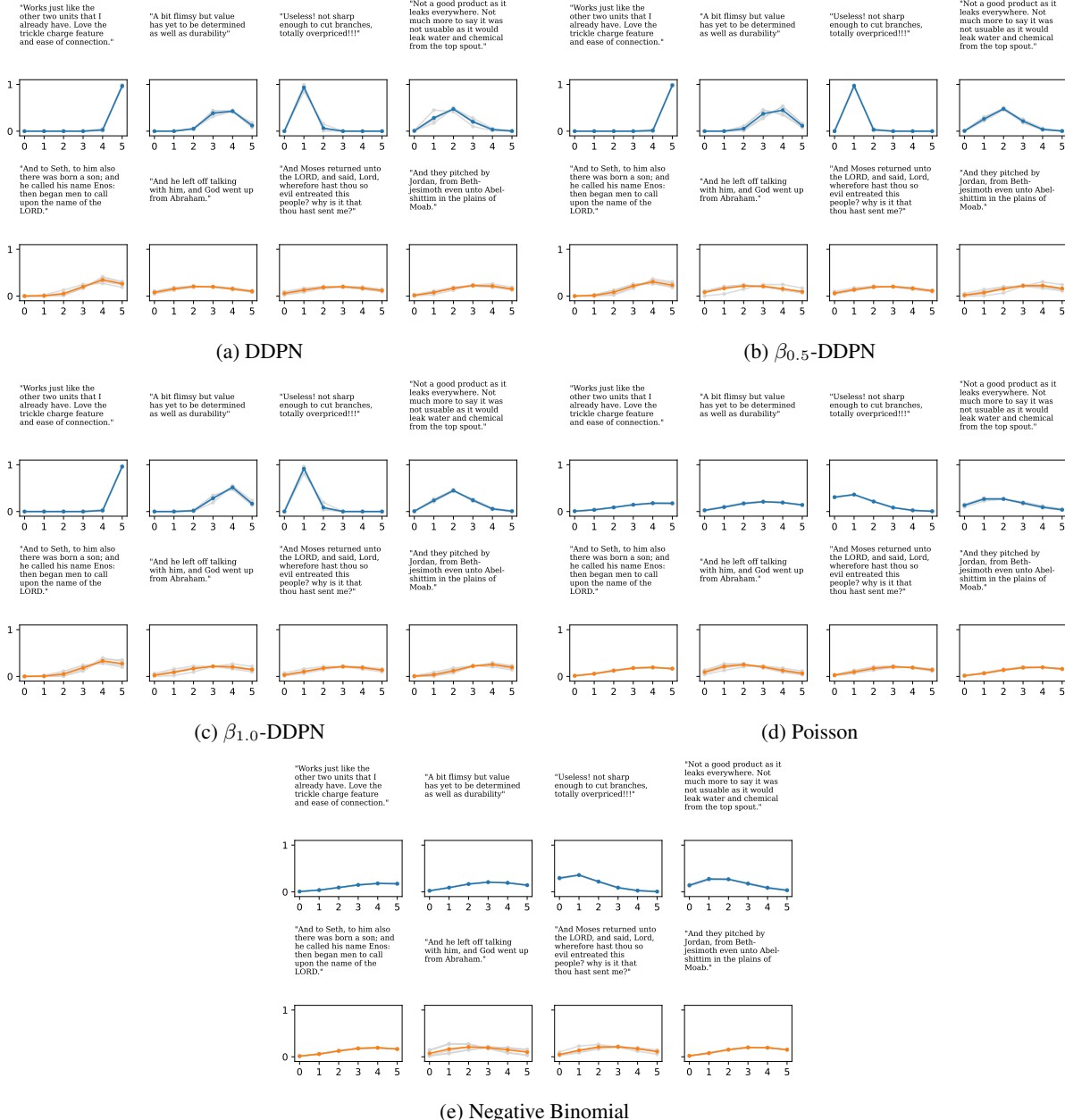

Figure 30: "Discrete" ensemble predictions on in-distribution (ID) vs. out-of-distribution (OOD) examples. Individual ensemble members were trained on `Amazon Reviews` and then fed verses from the King James Version of the Bible. Note the desirable behavior of DDPN (and $\beta$-DDPN) ensembles, which produce concentrated probabilities on ID examples but appear to revert to the optimal constant solution (OCS) when faced with OOD examples (Kang et al., 2023). Meanwhile, the predictive variance exhibits little difference between ID and OOD examples for the Poisson and Negative Binomial ensembles.

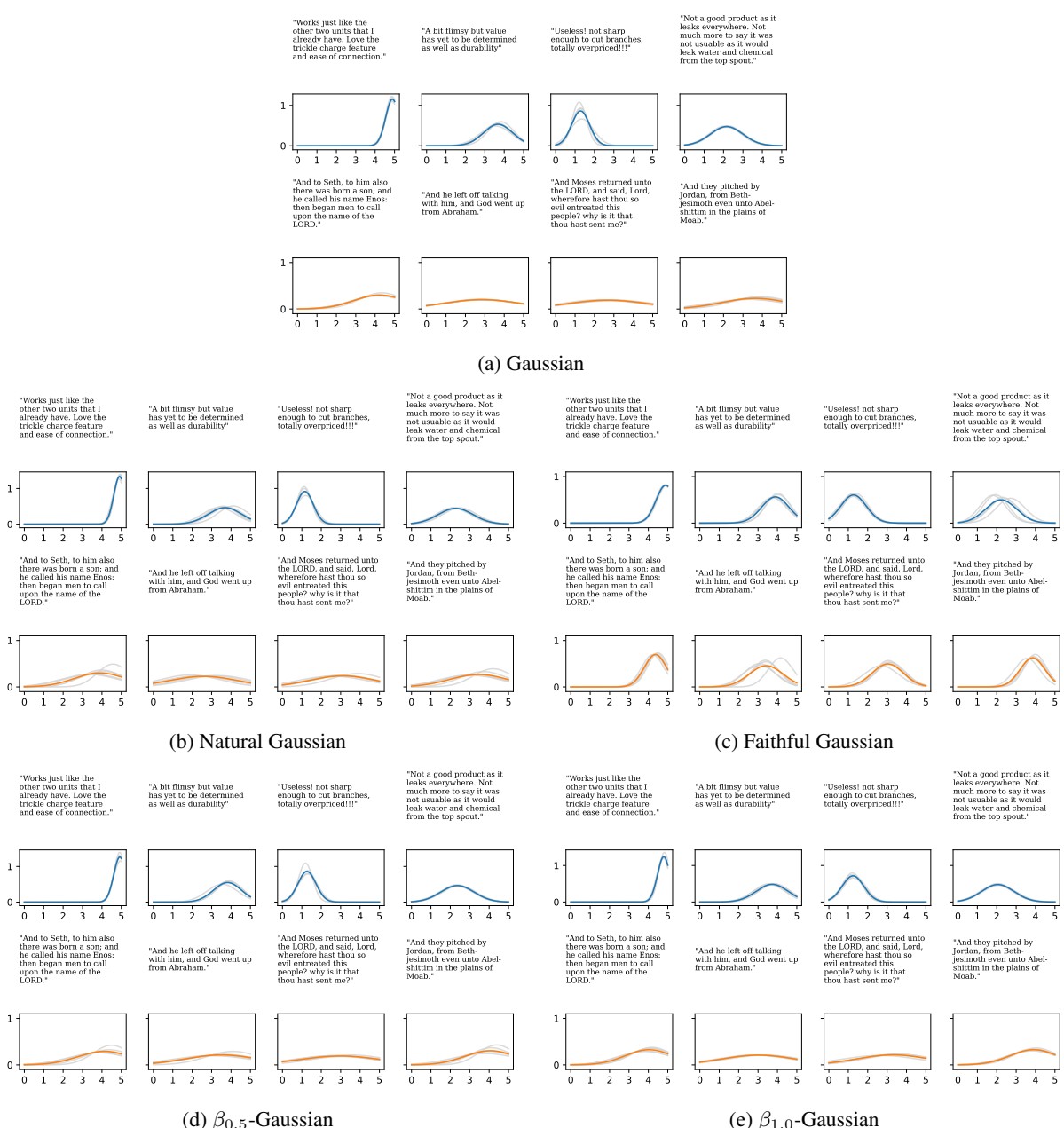

Figure 31: "Continuous" ensemble predictions on in-distribution (ID) vs. out-of-distribution (OOD) examples. Individual ensemble members were trained on `Amazon Reviews` and then fed verses from the King James Version of the Bible.

### D.3. Example Point Cloud from `Inventory`

In Figure 32, we provide an example point cloud from the `Inventory` dataset used in the experiments of Section 4.3. Further examples can be viewed in Jenkins et al. (2023).

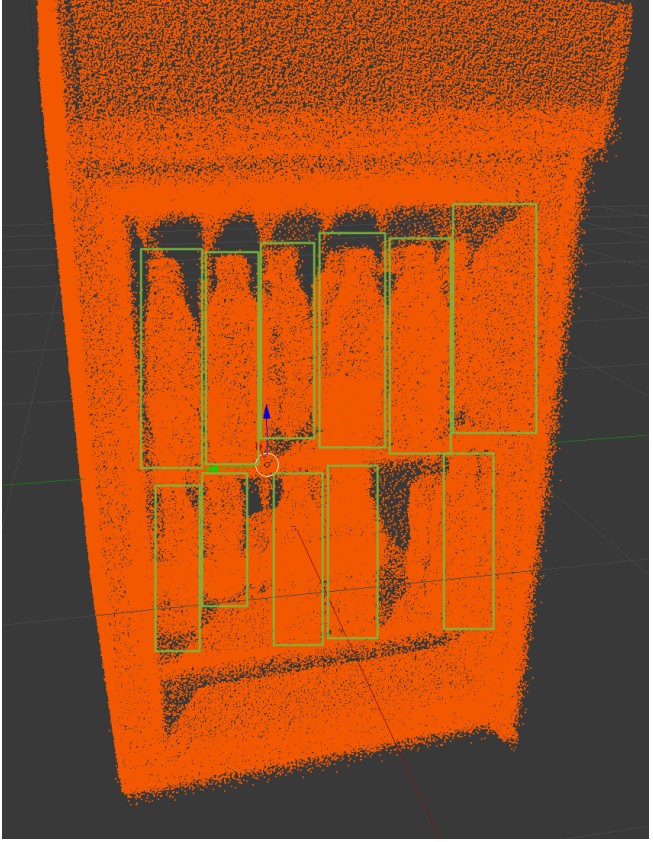

Figure 32: Example point cloud from `Inventory`. Each green box represents an inventory slot which is segmented into a point beam (see Jenkins et al. (2023) for details and further examples). Models predict the product count within each point beam.

