# OpenReview forum: "Fully Heteroscedastic Count Regression with Deep Double Poisson Networks"
_ICML.cc/2025/Conference — ICML 2025 poster_

### Official Review · Reviewer_bN97 · 2025-03-13

**Overall Recommendation:** 2

**Summary:**

The paper introduces the Deep Double Poisson Network (DDPN), a novel neural network model for count regression that provides accurate input-conditional uncertainty quantification. The main conceptual idea is that DDPN extends deep ensembles to count regression by using the Double Poisson distribution, which allows for heteroscedastic variance in count data. This flexibility enables improved estimation of aleatoric uncertainty (inherent variability in data) and, consequently, better epistemic uncertainty (model uncertainty) estimation. The paper proves that DDPN exhibits properties similar to heteroscedastic Gaussian models. The authors introduce a loss modification to control the learnable loss attenuation mechanism, allowing for more precise uncertainty calibration. Experiments on diverse datasets show that DDPN outperforms existing count regression baselines in accuracy, calibration, and out-of-distribution detection.

## update after rebuttal
I acknowledge that the authors have improved the proofs (my point 1.) but do not provide a strong argument for point 2. I will increase my score to 2, but still think the work does not reach the acceptance bar.

**Claims And Evidence:**

Overall, the claims made in the submission are supported by clear evidence, but I found that some theoretical claims are not supported by convincing mathematical arguments (see below).

**Essential References Not Discussed:**

Not that I am aware of.

**Experimental Designs Or Analyses:**

Owing to the previous flaws, I did not check the soundness of the experimental analyses.

**Methods And Evaluation Criteria:**

Yes.

**Other Comments Or Suggestions:**

- The second line of the displayed equation in app C.6 is hard to parse: adding parentheses to the rhs 2nd sum would help.

- There are a few (math) typos that could easily be fixed.

**Other Strengths And Weaknesses:**

I identify a weakness in Proposition 3.1, which claims that DDPN regressors are fully heteroscedastic. In reality, the proposition is derived under the moment approximations proposed by Efron (1986), where the first two moments are approximated as \mu and \mu / \gamma. Given this approximation, full heteroscedasticity is unsurprising. It would be more meaningful to establish this result without relying on Efron’s approximation. I suspect that the exact Double Poisson distribution is inherently fully heteroscedastic, and proving this directly would be a more valuable contribution.

**Questions For Authors:**

Given the identified flaws and weaknesses, I would likely revise my evaluation if the authors:
1. Provide convincing proofs that correct the identified issues.
2. Establish a stronger version of full heteroscedasticity for Double Poisson regressors without relying on moment approximations.

**Relation To Broader Scientific Literature:**

The paper builds on prior work in deep ensembles for uncertainty estimation in regression, extending heteroscedastic modeling from Gaussian outputs to count data using the Double Poisson distribution, addressing a gap in discrete uncertainty modeling and improving epistemic uncertainty quantification.

**Theoretical Claims:**

I checked the theoretical claims and found significant flaws. Some may be fixable, but others appear more critical.

- One issue concerns Proposition 3.3: the stated convergence to 0 does not seem valid under the proposed definition of the learnable attenuation loss function (Definition 3.2). A monotonically increasing function does not necessarily tend to infinity, just as a monotonically decreasing function does not necessarily tend to 0—both cases can have a constant asymptote. This flaw undermines the argument in the proof of Proposition 3.3.

- Another issue concerns the derivation of the DDPN objective. The derivation in Appendix A.1 omits the normalizing constant of the Double Poisson (DP) distribution, denoted c(\mu, \gamma) at the beginning of Section 3. Since c(\mu, \gamma) is not constant with respect to (\mu, \gamma), the stated objective does not properly learn the parameters of a DDPN.

- Additionally, most equations in Appendix A.1 fail to hold because the maxima and minima do not align due to omitted constants between successive lines. Using \arg\max and \arg\min would provide more precision. Also, the parameterization of the network f_\Theta(x_i) is inconsistent: while a log link function is used in the main text, this log transformation is omitted in the first line of Appendix A.1.

- Finally, it is unclear why the DDPN objective loss in Equation (1) of Section 3.1 does not include a summation over all training examples i = 1, \dots, N.

---

> ### Author Rebuttal · Authors · 2025-03-31
>
> We thank the reviewer for the thoughtful feedback.
>
>
> ## Monotonicity vs tend to infinity
>
> We agree with this remark and propose to change Def. 3.2 to:
>
> >  where $\lim_{\hat{\phi} \to \infty} d(\hat{\phi}) = \infty$ and $\lim_{\hat{\phi} \to \infty} a(\hat{\phi}) = 0 $
>
> Fortunately, the proof in Appendix C.3 holds under this new definition, as these limits are in fact used to show that the residual error tends towards 0 (lines 1108-1109). With this change, our proof of Prop. 3.4 also remains valid (since logx tends to infinity and 1/x tends to zero).
>
> ## Normalizing constant
>
> To simplify our objective, we followed previous work (see below) and assumed $c(\mu, \gamma) = 1$. This facilitates easier differentiation. We will make this more clear in App. A.1.
>
> - See Fact 1 (Eqn. 2.10) of [Efron, B. "Double exponential families and their use in generalized linear regression." Journal of the American Statistical Association. 1986]
> - Follow-up work has also set $c(\mu, \gamma)=1$ [Chow, N., and David Steenhard.  2009. "A flexible count data regression model using SAS Proc nlmixed”]
>
> ## Max/Min in Appendix 1
>
> We propose two changes to the derivation of our objective to increase clarity and align with convention:
>
> 1. Replace max/min with argmax/argmin
> 2. Clarify that we maximize over $N$ training examples (see below) and derive the per-instance loss defined in Equation 1 (see discussion below)
>
> The NLL becomes: $\arg\min_{\mu_i, \gamma_i} [ -\sum_{i=1}^N \log p(y_i | \mu_i, \gamma_i)]$.
>
> ##  Lack of Log link in Appendix A.1
>
> In App. A.1 we derive the training objective. In contrast, Section 3 describes how it can be used to train DDPN. The connection between the two is trivial (exponentiate the log link to evaluate Eq. 1) and is stated in Footnote 1 (pg. 4).
>
> ## Lack of Summation in Equation 1
>
> Eq. 1 expresses the loss for a single training example, $x_i$. We state in lines 183-184 that the loss is:
> > averaged across all prediction / target tuples…in the dataset
>
> To be more explicit, we propose to change $\mathcal{L}$ to $\mathcal{L}_i$.
>
> ## Prop. 3.1: DDPN regressors and full heteroscedasticity
>
> In line with prior work, we use Efron's approximations for the first two moments in the proof of Prop. 3.1. We propose to revise this proposition: *With mild assumptions*, DDPN regressors are fully heteroscedastic (where we assume that Efron's approximations hold).
>
> How good are Efron's approximations? We introduce the concept of moment-deviation functions (MDFs) to assess this theoretically:
>
> Let $Q$ be a family of distributions parametrized by $\psi\in\mathbb{R}^d$. Suppose we are given $\hat{\psi_n}:\mathbb{R}^n\to\mathbb{R}^d$, which outputs $\hat{\psi_n}(\boldsymbol{\mu})$ s.t. the first $n$ moments of $Z\sim Q_{\hat{\psi_n}(\boldsymbol{\mu})}$ are nearly equal to target moments $\boldsymbol{\mu}=(\mu_1,...,\mu_n)$. Then for any pair $(Q,\hat{\psi_n})$, {$ \varepsilon_i:\mathbb{R}^n\rightarrow\mathbb{R}$} for $i=1..n$ are moment-deviation functions if, for any valid $\boldsymbol{\mu}$, if $Z\sim Q_{\hat{\psi_n}(\boldsymbol{\mu})}$, we have $|\langle Z^i\rangle -\mu_i|\leq\varepsilon_i(\boldsymbol{\mu})$ for all $1\leq i\leq n$.
>
> We focus on n=2, (error for the mean/variance). If we can pick $\hat{\psi}$ s.t. $\varepsilon_1,\varepsilon_2$ are small, $Q$ is flexible, as there are parameters that can roughly achieve any mean and variance. In the Gaussian case ($\mathcal{N},\mathbb{I}$), we have $\varepsilon_1 =\varepsilon_2=0$.
>
> ### Proposition
>
> Let $DP$ denote the Double Poisson family. Set $\hat{\psi_2}(\mu_0, \sigma_0^2) = (\mu_0,\frac{\mu_0}{\sigma_0^2})$. Letting $\gamma_0=\frac{\mu_0}{\sigma_0^2}$, the MDFs for $(DP,\hat{\psi_2})$ are:
>     \begin{align*}
>         \varepsilon_1(\mu_0,\sigma_0^2) &= \left|\frac{\sum_{y=0}^{\infty}s(\mu_0, \gamma_0, y)(y - \mu_0)}{\sum_{y=0}^{\infty}s(\mu_0,\gamma_0,y)}\right|  \\\\
>         \varepsilon_2(\mu_0, \sigma_0^2)&=\left|\frac{d(\mu_0,\gamma_0)\gamma_0^{\frac{1}{2}}\sum_{y=0}^{\infty}s(\mu_0,\gamma_0,y)-\gamma_0(\sum_{y=0}^{\infty}s(\mu_0, \gamma_0,y)(y-\mu_0))^2}{\gamma_0(\sum_{y=0}^{\infty}s(\mu_0,\gamma_0,y))^2}\right|
>     \end{align*}
>     where: $h(z)=\frac{e^{-z} z^z}{z!}, r(\mu,\gamma,z)=\gamma(z-\mu +z\log\mu -z\log z), s(\mu,\gamma,z)=h(z)\exp(r(\mu,\gamma,z)),$ and $d(\mu,\gamma)=\gamma^{-1/2}\left[\sum_{y=0}^{\infty}s(\mu, \gamma, y)(\gamma(y-\mu)^2-y)+\sum_{y=0}^{\infty}s(\mu,\gamma,y)(y-\mu)\right]$.
>
> If desired, we can provide the proof to this proposition in the follow-up response. We plot the error incurred via Efron’s estimates on a grid of target means and variances, using 100th partial sums (https://anonymous.4open.science/r/ddpn-651F/deep_uncertainty/figures/artifacts/epsilon_1.png). To see epsilon_2, change the filepath to epsilon_2.png. Except for the case of small μ, high σ², the error is essentially zero. Thus, in most settings we can treat DDPN as fully-heteroscedastic. Empirically, DDPN produces flexible, well-fit distributions (Fig. 3/4, Table 2).

---

> > ### Comment · Reviewer_bN97 · 2025-04-02
> >
> > ### Monotonicity vs tend to infinity
> > I agree that the proposed change to Def. 3.2 make the proof in Appendix C.3 now possible.
> >
> > ### Normalizing constant
> > The authors replied:
> > > assumed $c(\mu, \gamma) = 1$.
> >
> > I do not think this assumption was made clear anywhere in the submitted paper; it could only be discovered by checking the earlier work by Efron.
> >
> > ### Max/Min in Appendix 1
> > > We propose two changes to the derivation of our objective to increase clarity and align with convention:
> >
> > Replacing max/min with argmax/argmin is not just a question of clarity or convention: the proof is simply wrong without.
> >
> > ### Lack of Log link in Appendix A.1
> > I understand that the connection with and without log link is trivial. But as I noticed, the parameterization of the network $f_\Theta(x_i)$ is inconsistent between main text and supplementary.
> >
> > ### Lack of Summation in Equation 1
> > I agree that changing $\mathcal{L}$ to $\mathcal{L}_i$ helps with clarity.
> >
> > ### Prop. 3.1: DDPN regressors and full heteroscedasticity
> > Introducing moment-deviation functions to assess the error of Efron’s approximation is a nice idea. However, I still believe that demonstrating full heteroscedasticity for the *exact* Double Poisson distribution should be the primary goal. After all, full heteroscedasticity simply means that, for any fixed mean, the variance can span the entire interval $(0, \infty)$. I genuinely think this is an attainable property for the *exact* Double Poisson distribution.
> >
> > ## Score revision
> > I acknowledge that the authors have improved the proofs (my point 1.) but do not provide a strong argument for point 2. I will increase my score to 2, but still think the work does not reach the acceptance bar.

---

> > > ### Author Response · Authors · 2025-04-02
> > >
> > > We appreciate the additional thoughtful comments from the reviewer. As discussed, we will make all of the proposed improvements to the proofs (point 1) in the camera ready manuscript. With respect to point 2, we will include a discussion of moment-deviation functions (and the quality of Efron's approximations) in the appendix.

---

### Official Review · Reviewer_68NH · 2025-03-14

**Overall Recommendation:** 3

**Summary:**

The work introduces deep double poisson networks for the count regression problem. The proposed approach can quantify both the aleatoric and epistemic uncertainty with ensemble. Also, double poisson network allows unrestricted variance to model discrete count data, and can show robustness to outiers. Authors carry out experiments where the approach performs better in terms of calibration, out-of-distribution detection, and accuracy.

**Claims And Evidence:**

- The authors claim that the proposed deep double poisson network can perform well on the count regression task. The claims are empirically validated through experiments on benchmark datasets and some baseline methods.

**Essential References Not Discussed:**

The work Natural Posterior Network: Deep Bayesian Uncertainty for Exponential Family Distributions [Bertrand Charpentier, Oliver Borchert, Daniel Zügner, Simon Geisler, Stephan Günnemann]) introduces a general evidential approach that can be effective for a wide range of problems including uncertainty-aware classification, uncertainty-aware regression, and uncertainty-aware count regression. Discussion and comparison with the work could be beneficial.

**Experimental Designs Or Analyses:**

The experimental design looks sound.

**Methods And Evaluation Criteria:**

The methods and evaluation criteria look reasonable. The authors consider discrete count regression problem, and look at the different metrics, evaluating the method along different dimensions including OOD detection, calibration, and accuracy.

**Other Comments Or Suggestions:**

- Figures, labels and captions can be better presented. Many labels/legend texts are too small and not clearly legible. Also, the captions are too long and could be shortened for a better read.

**Other Strengths And Weaknesses:**

Strengths
- The paper is easy to follow and I found it to be a pleasant read.
- The work introduces deep double possion network which seems to be effective in discrete count regression based on the experimental results The authors show the robustness to outliers of the proposed approach. Also, the approach performs well in terms of accuracy, calibration, and ood detection.
Weaknesses:
- Beyond ensembling, there are other approaches (e.g., Bayesian neural networks, evidential approaches, dropout-based uncertainties). While the authors compare with standard Poisson, NB, and Gaussian-based heteroscedastic networks, a thorough comparison can help better illustrate the effectiveness of the approach.

**Questions For Authors:**

- How does the work perform compared to natural posterior networks and the presented baselines on the benchmark Bike Sharing dataset (Natural Posterior Network: Deep Bayesian Uncertainty for Exponential Family Distributions [Bertrand Charpentier, Oliver Borchert, Daniel Zügner, Simon Geisler, Stephan Günnemann]) ?

**Relation To Broader Scientific Literature:**

The work is likely to have limited impact to a narrow subfield in the scientific community.

**Theoretical Claims:**

- Authors present some theoretical claims,  but these seem to be derived from standard double poisson networks.

---

> ### Author Rebuttal · Authors · 2025-03-31
>
> We appreciate the reviewer’s thoughtful comments and helpful feedback.
>
> ## Comparison to the Natural Posterior Network
> We have followed the official repository to download the `bike-sharing` dataset file and pre-process exactly as used in the paper reviewer mentioned. For training, the paper mentioned they perform training after the grid search in the space `[1e−2, 5e−4]`, since the search step is not specified, we took the log-scale space step as below: [0.01, 0.005623, 0.003162, 0.001778, 0.001, 0.000708, 0.0005], and found the lr as: 0.003162 in our model’s configuration. Following the exact same settings, we conducted evaluation on 5 rounds of training and report the mean and standard deviation of our model’s performance on RMSE, the results are:
>
>
> | Method         | RMSE         |
> |----------------|--------------|
> | Dropout-N      | 70.20 ± 1.30 |
> | Ensemble-N     | 48.02 ± 2.78 |
> | EvReg-N        | 49.58 ± 1.51 |
> | NatPN-N        | 49.85 ± 1.38 |
> | Dropout-Poi    | 66.57 ± 4.61 |
> | Ensemble-Poi   | 48.22 ± 2.06 |
> | NatPN-Poi      | 51.79 ± 0.78 |
> | **DDPN (ours)**| **47.87 ± 0.42** |
>
> ## How does DDPN compare to other uncertainty methods?
>
> In short, the objective function proposed in Equation 1 enables the network to capture aleatoric uncertainty over count data. We show how this can be combined with Deep Ensembles to better capture epistemic uncertainty. Table 2 shows this combination is effective. DDPN is presented in our paper in terms of maximum likelihood + ensembles for 1) simplicity, 2) effectiveness, and 3) likelihood of community adoption.  DDPN can easily be combined with other UQ methods.
>
>
> ### Bayesian Neural Networks
>
> One could put a prior over the weights of the network (ideally an isotropic Gaussian prior). Let $\theta$ denote the neural network parameters, $f_\Theta(x_i)$, and $\mathcal{D}$ denote the training dataset. The log posterior is:
>
> $\log p(\theta | \mathcal{D})) = \log \frac{1}{Z} + \log p(\mathcal{D} | \theta) + \log p(\theta)$
>
> where $\frac{1}{Z}$ is the normalizing partition function and is usually dropped during inference.
>
> Fortunately, the negative log likelihood, $-\log p(\mathcal{D} | \theta)$ is already defined in Equation 1 of our paper, and the log gaussian prior is easy to compute, $\log p(\theta) = -\frac{1}{2} \log (2\pi \sigma_0^2) - \frac{1}{2\sigma_0^2} (\theta - \mu_0)^2$, where the prior hyperparameters are $\mu_0$ and $\sigma_0^2$.
>
> Then one could choose the preferred inference algorithm: MAP, HMC, SGLD etc… and estimate the posterior.
>
> Empirically, Bayesian neural networks can outperform Deep Ensembles when using high fidelity inference algorithms such as Hamiltonian Monte Carlo. However, in practice MCMC-based inference is impractical and DEs often outperform less exact inference methods (i.e., SGLD, Variational Inference etc…) [Izmailov et al. What Are Bayesian Neural Network Posteriors Really Like? ICML’21]. We suspect the same results hold for DDPNs.
>
> Moreover, many recent works have directly connected Deep Ensembles to Bayesian Inference by showing that DEs are a coarse approximation of the posterior, sampled and multiple modes with no local uncertainty [Fort et al. Deep Ensembles: A Loss Landscape Perspective. 2019][Wilson and Izmailov. Bayesian Deep Learning and a Probabilistic Perspective of Generalization. NeurIPS’20].
>
> The effectiveness, simplicity and attractive theoretical properties of DEs motivated our decisions to use them in our experiments.
>
> ### Evidential Approaches
>
> DDPN could also be easily applied with evidential regression techniques [Amini et al. Deep Evidential Regression. NeurIPS’20]. Because DDPN uses a likelihood function during training, one would simply have to specify evidential priors over the parameters of the DDPN, $p(\mu)$ for the mean and $p(\gamma)$ for the inverse dispersion. Then, the network would be trained to predict the parameters of the higher-order evidential distribution.
>
> ### Laplace Approximation (LA)
>
> This is perhaps the easiest since LA is typically a post-hoc technique. One would train the DDPN in the standard way described in our paper. Then, one could apply any of the post-hoc second-order, covariance approximation methods described in [Daxberger et al. Laplace Redux – Effortless Bayesian Deep Learning. NeurIPS’21]
>
> ### Dropout-based uncertainty
>
> Monte Carlo dropout estimates epistemic uncertainty by randomly dropping out weights at test time and approximates the Bayesian posterior.
>
> DDPN can easily be combined with MC dropout by 1) training the single-member DDPN to convergence, and 2) applying the MC dropout procedure with $T$ different forward passes through the dropped out model [Gal and Ghahramani. Dropout as a Bayesian Approximation: Representing Model Uncertainty in Deep Learning. ICML’16]
>
> However, [Lakshminarayanan et al. Simple and Scalable Predictive Uncertainty Estimation using Deep Ensembles. NeurIPS’17] show that MC dropout is clearly inferior to DEs.

---

> > ### Comment · Reviewer_68NH · 2025-04-03
> >
> > The authors have addressed my comment and I vote to keep my original score of weak accept.

---

### Official Review · Reviewer_MbuP · 2025-03-14

**Overall Recommendation:** 4

**Summary:**

In this paper, the authors consider the problem of estimating heteroscedastic uncertainty within the context of counting tasks, where the final outputs should represent positive integer numbers. While many successful solutions have been proposed for heteroscedastic uncertainty in general (real-valued) regression tasks, this is not the case for counting, as it requires a different parametrization of the output distribution. Earlier solutions for the counting setting, such as the Poisson distribution, suffer from restricted heteroscedastic variance, meaning that the parameter defining the mean value of the distribution significantly restricts the possible predicted variance. In this paper, the authors propose using the Double Poisson distribution for counting tasks and prove that it resolves the issue of the former method, namely, it has unrestricted variance. Additionally, they demonstrate that the proposed loss has the property of adaptive loss attenuation, which lowers the impact of outlier points during training. Finally, they propose a way to make this attenuation controllable through $\beta$-DDPN. The effectiveness of the proposed method is demonstrated on several datasets from different domains, showing that the proposed parametrization outperforms other parametrizations for counting tasks in terms of uncertainty quality (calibration) and accuracy.

**Claims And Evidence:**

The authors present their claims and contributions in a clear manner while also supporting them with both theoretical (for example, proving that the proposed DDPN regressors are fully heteroscedastic) and experimental results.

**Essential References Not Discussed:**

No, there are no critical references missing in the paper.

**Experimental Designs Or Analyses:**

The experimental design and analysis are adequate and rigorous, with no issues.

**Methods And Evaluation Criteria:**

The authors primarily compare against other loss-based heteroscedastic approaches on various counting tasks, clearly demonstrating the effectiveness of the proposed approaches in the discussed counting setups.

**Other Comments Or Suggestions:**

N/A

**Other Strengths And Weaknesses:**

In short, the major Strengths of the paper are:

* The paper clearly discusses the problem of heteroscedastic uncertainty estimation in the counting context, its existing problems, and solutions.
* The proposed method is clear, easy to implement, and demonstrates good performance on a number of different tasks.
* In contrast to many other uncertainty methods, it does not require a significant increase in computation/memory during inference while still producing high-quality uncertainty estimations.

One of the potential Weaknesses:

* The paper mostly compares the method against other loss-based uncertainty methods. Introducing additional uncertainty approaches, such as ensembling methods (Deep Ensembles, Batch Ensembles, etc.), could be beneficial.

**Questions For Authors:**

N/A

**Relation To Broader Scientific Literature:**

The paper clearly positions itself within the existing literature by thoroughly discussing prior work on heteroscedastic uncertainty estimation, particularly in regression and counting tasks. It provides sufficient detail on previous parameterizations, such as Poisson and Negative Binomial, highlighting their limitations and demonstrating how the proposed DDPN framework overcomes these constraints.

**Theoretical Claims:**

The main theoretical contributions of the paper could be considered Propositions 2.3 and 3.1, which prove that previously proposed parameterizations, such as Poisson and Negative Binomial, are not fully heteroscedastic, while DDPN is. The proposed proof appears to be correct and valid, with no observable issues.

---

> ### Author Rebuttal · Authors · 2025-03-31
>
> We appreciate the reviewer’s thoughtful comments.
>
> ##  Introducing additional uncertainty approaches, such as ensembling methods (Deep Ensembles, Batch Ensembles, etc.), could be beneficial.
>
> An important aspect of our work is the interplay between DDPN and Deep Ensembles. We demonstrate this connection throughout the paper. In Section 3.4 we show how individual DDPNs can be combined as ensembles. Then, in the bottom half of Table 2 we present results with ensemble DDPNs. Table 2 demonstrates that learning ensembles of DDPNs improves accuracy **and** the quality of predictive uncertainty.
>
> We suspect that similar results will hold for BatchEnsembles, as this type of ensemble just changes how the weights of each member are derived: combining slow shared weights and fast, independent rank one weights. The principles we propose in this paper could easily be applied to other types of ensembles. We leave the validation of this hypothesis to future work.
>
> Finally, in the discussion with Reviewer 68NH, we discuss how DDPN can be combined with other UQ methods such as Bayesian Neural Nets, Evidential Methods, Laplace Approximation and Monte Carlo Dropout.

---

### Official Review · Reviewer_AhTw · 2025-03-25

**Overall Recommendation:** 4

**Summary:**

The paper focuses on outputting distributions for non-negative integer predictions (i.e., count data). To do so, the paper has a model output the parameters for a Double Poisson distribution, which admits separate mean and variance parameterizations. Then the paper further utilize ensembles to include epistemic model uncertainty. Results comparing against other predictive distributions, such as a typical Gaussian, show that the proposed predictive distribution outperforms on count tasks.

**Claims And Evidence:**

Yes, the paper is quite clear on the proposed approach, the different uncertainties involved (e.g., the diff between aleatoric and epistemic, which is often confused or muddled), and the experimental setup. The reasoning for using the Double Poisson instead of a regular Poisson is clear and backed by both theory and empirical evidence.

**Essential References Not Discussed:**

No

**Experimental Designs Or Analyses:**

Yes, the experimental setup, including datasets, metrics, and baselines, are appropriate and expected for the type of approach being proposed. E.g., accuracy and a proper scoring rule is an ideal combination (often, the latter is missed), and the baselines are appropriately other predictive distributions and do not conflate that with other modeling choices.

**Methods And Evaluation Criteria:**

Yes, the proposed Double Poisson makes sense for count data and for the goal of heteroscedastic variance (i.e., per-example variance controlled by the model). The evaluation datasets are fine and varied, and the baselines being other predictive distributions is appropriate and expected.

**Other Comments Or Suggestions:**

None

**Other Strengths And Weaknesses:**

This paper's novelty is in focusing on count data and using the less common Double Poisson for full heteroscedasticity. It's not a surprisingly result and is fairly straightforward, but it's a useful paper to have in the literature. In particular, I'm pleased by the way in which the paper is very clear on the various uncertainty concepts and does not confuse or conflate any terms.

I am assigning a "4: Accept" to mean that it's a solid paper; a 5 would be for an exceptionally exciting result, such as showing that this pushes on SoTA in some current frontier model.

**Questions For Authors:**

None

**Relation To Broader Scientific Literature:**

This fits well within the broader literature on uncertainty quantification. Accordingly, the appropriate papers are generally referenced. Most existing work has focused on categorical or continuous problems. This paper's novelty is in focusing on count data and using the less common Double Poisson for full heteroscedasticity.

**Theoretical Claims:**

The formal definitions of the distributions (e.g., 2.1, 2.2, 3.2) and the propositions appear to be correct. In general, the approach is straightforward: have a model output the parameters of the Double Poisson distribution and then optimize that distributions NLL; this is generally the same type of approach used in modern models, just with a different distributional family.

---

> ### Author Rebuttal · Authors · 2025-03-31
>
> We sincerely thank the reviewer for the recognition of our work.

---

### Decision · Program_Chairs · 2025-05-01

**Decision:**

Accept (poster)

**Comment:**

This work considers the problem of non-negative integer prediction (count regression) with heteroscedastic uncertainty by fitting neural networks to the parameters of a double Poisson distribution. The resulting regressor is shown to be fully heteroscedastic with learnable loss attenuation, a property also enjoyed by Gaussian heteroscedastic regressors. To get clearer uncertainty quantification, ensembling is performed. These models are shown to be competitive through a variety of experiments. Reviewers were generally positive about the work; while not necessarily found to be paradigm-shifting, reviewers agreed with the soundness of the model, appreciated the comprehensive analysis, and the variety of experimental findings.

The most major concerns identified during the review period have been addressed in the author rebuttal, however, some concerns remain regarding the proof of heteroscedasticity, as this was found to hold only under approximations for the moments. I agree that this is an inherent limitation of the work, but can appreciate that proving general heteroscedasticity in this case may be out of reach. I would suggest that the authors include the evidence provided in their rebuttal, but adjust their wording to acknowledge they only show "approximate heteroscedasticity" under the moment assumptions. This needs to be _very_ clear and precisely defined. This is not imposing "mild assumptions", this is showing that the approximate moments are unrestricted. With these inclusions, as well as those discussed throughout this review period, I believe this is is a good contribution to ICML.